# T cell migration requires ion and water influx to regulate actin polymerization

Leonard L. de Boer [1,2,5], Lesley Vanes[1], Serena Melgrati[1,2,6], Joshua Biggs O'May[1], Darryl Hayward[1,7], Paul C. Driscoll [1], Jason Day [3], Alexander Griffiths[4], Renata Magueta[4], Alexander Morrell [4], James I. MacRae [1], Robert Köchl[1,8] & Victor L. J. Tybulewicz [1] ✉

Migration of T cells is essential for their ability to mount immune responses. Chemokine-induced T cell migration requires WNK1, a kinase that regulates ion influx into the cell. However, it is not known why ion entry is necessary for T cell movement. Here we show that signaling from the chemokine receptor CCR7 leads to activation of WNK1 and its downstream pathway at the leading edge of migrating CD4[+] T cells, resulting in ion influx and water entry by osmosis. We propose that WNK1-induced water entry is required to swell the membrane at the leading edge, generating space into which actin filaments can polymerize, thereby facilitating forward movement of the cell. Given the broad expression of WNK1 pathway proteins, our study suggests that ion and water influx are likely to be essential for migration in many cell types, including leukocytes and metastatic tumor cells.

Migration is an essential aspect of T cell physiology. Naive T cells migrate between lymphoid organs in search of antigen, a process that is essential for adaptive immune responses[1]. T cells exit the bloodstream and enter lymph nodes in response to the chemokine CCL21 presented on blood vessel endothelium. CCL21, acting through the CCR7 receptor on T cells, triggers integrin-mediated adhesion between the T cell and the endothelium. T cells then transmigrate through the endothelium and enter the parenchyma of the lymph node. Here they continue to migrate rapidly in response to CCL19 and CCL21 signaling through CCR7, now predominantly using an integrin-independent mode of migration to scan antigen-presenting cells for the presence of cognate antigen, a process that is critical for an efficient immune response[2].

Chemokine-induced migration of T cells is characterized by polarization of the cell, with the formation of a pseudopodium at the leading edge of the migrating cell. This structure contains a dense and highly dynamic network of branched actin filaments which is essential for migration[3]. Chemokine receptor signaling at the leading edge through phosphatidylinositol 3-kinase (PI3K) results in activation of RAC1, RAC2 and CDC42 GTPases which promote actin polymerization and the branching of actin filaments. Actin polymerization pushing at the leading edge in turn provides the motor that drives the plasma membrane forward, resulting in migration.

WNK1 is a member of the WNK-family of serine/threonine protein kinases that regulate ion homeostasis in kidney epithelial cells where they promote uptake of $Na^+$, $K^+$ and $Cl^-$ ions from urine[4]. When activated, WNK1 phosphorylates and activates two related kinases OXSR1 and STK39, which in turn phosphorylate members of the SLC12A-family of electroneutral ion co-transporters, altering their activity. For example, phosphorylation of SLC12A2 (NKCC1) causes influx of $Na^+$, $K^+$ and $Cl^-$ ions, whereas phosphorylation of SLC12A6 (KCC3) blocks efflux of $K^+$ and $Cl^-$ ions through the transporter[5]. Thus, signaling from WNK1 results in a net influx of ions.

[1]The Francis Crick Institute, London NW1 1AT, UK. [2]Department of Immunology and Inflammation, Imperial College London, London W12 0NN, UK. [3]Department of Earth Sciences, University of Cambridge, Cambridge CB2 3EQ, UK. [4]London Metallomics Facility, Research Management & Innovation Directorate, King's College London, London SE1 1UL, UK. [5]Present address: Science for Life Laboratory, Department of Women's and Children's Health, Karolinska Institutet, 171 65 Stockholm, Sweden. [6]Present address: Institute for Research in Biomedicine, Università della Svizzera Italiana, Bellinzona, Switzerland. [7]Present address: GSK, Stevenage SG1 2NY, UK. [8]Present address: Kings College London, London SE1 9RT, UK. ✉e-mail: Victor.T@crick.ac.uk

We previously demonstrated that WNK1, the only WNK-family member expressed in mouse T cells, is required for the homing of CD4[+] and CD8[+] T cells to lymph nodes and spleen[6]. Furthermore, we showed that WNK1-deficient CD4[+] T cells migrated more slowly in vivo in lymph node parenchyma[6]. WNK1-deficient T cells show increased integrin-mediated adhesion, which could have accounted for this reduced in vivo migration speed. However, blocking integrin adhesion further reduced in vivo migration speed, demonstrating that hyperadhesion was not the cause of reduced migration. We also showed that WNK1-deficient T cells have defective chemokine-induced migration in vitro[6]. We demonstrated that stimulation of CD4[+] T cells with CCL21 results in rapid WNK1-dependent phosphorylation of OXSR1, which also required PI3K, indicating that CCR7 transduces signals via PI3K leading to WNK1 activation. Furthermore, CCL21-induced migration was partially inhibited in CD4[+] T cells with mutations in *Oxsr1* and *Slc12a2* or treated with bumetanide (SLC12A2i), an SLC12A2 inhibitor[6]. Taken together, these results lead to the surprising conclusion that CCR7 signaling via WNK1 results in ion influx via SLC12A2 which is required for T cell migration. Thus, a pathway regulating ion homeostasis plays a critical role in T cell movement. However, it remains unclear why ion influx is required for this process.

Here we demonstrate that WNK1 pathway proteins and their activities are polarized to the leading edge of migrating T cells. Furthermore, we show that chemokine receptor-induced activation of the WNK1 pathway leads to ion influx and to water influx, with the latter most likely occurring via AQP3. We provide evidence that water ingress is required for actin polymerization and migration. We propose that this chemokine receptor-induced water entry generates increased spacing between the plasma membrane and the underlying actin cytoskeleton at the leading edge, which could facilitate the extension of actin filaments, and hence forward migration.

## Results

### WNK1 pathway proteins are required for CCL21-induced migration of CD4[+] T cells

To understand how WNK1 pathway-regulated ion influx contributes to migration, we analyzed the functional requirement for the pathway proteins in T cells migrating in confinement, to model their migration within lymphoid tissue. Since constitutive loss of WNK1 in mice results in embryonic lethality[7] and T cell lineage-selective loss results in a strong developmental arrest in the thymus with the absence of any mature WNK1-deficient T cells[8], we chose to inducibly delete the *Wnk1* gene in mature naive T cells, as previously described[6]. We crossed mice bearing a loxP-flanked *Wnk1* allele (*Wnk1*[fl]) or a deletion of *Wnk1* (*Wnk1*[–]) to mice with a tamoxifen-inducible Cre recombinase under the control of the ubiquitously expressed ROSA26 promoter (*ROSA26*[CreERT2], RCE) to generate *Wnk1*[+/fl]RCE and *Wnk1*[fl/fl]RCE mice. Bone marrow from these animals was used to reconstitute the hematopoietic system of irradiated RAG1-deficient mice (Supplementary Fig. 1a). Subsequent treatment of these radiation chimeras with tamoxifen resulted in the generation of mice containing WNK1-expressing (*Wnk1*[+/–]RCE) and WNK1-deficient (*Wnk1*[–/–]RCE) T cells from which we isolated naive (CD25[–]CD44[–]) CD4[+] T cells for further study.

We imaged the CCL21-induced migration of these CD4[+] T cells under agarose on dishes coated with the integrin ligand ICAM-1. We found that in the absence of WNK1, T cells migrated more slowly (Supplementary Movie 1, Supplementary Fig. 2a, b, Fig. 1a). Moreover, whereas WNK1-expressing cells had a typical extended polarized morphology of migrating cells, WNK1-deficient cells showed no obvious polarization as measured by increased circularity (Supplementary Movie 1, Supplementary Fig. 2a–c). To evaluate the function of the WNK1 substrates OXSR1 and STK39 in T cell migration, we generated T cells mutated for both kinases, to account for any potential redundancy between them (Supplementary Fig. 1b). Using a

tamoxifen-inducible ablation of *Oxsr1*, we found that CD4[+] T cells deficient in OXSR1 and expressing a mutant STK39-T243A that cannot be phosphorylated and activated by WNK1[9] (*Oxsr1*[–/–]*Stk39*[T243A/T243A]RCE) also migrated more slowly in response to CCL21 in the same under agarose migration assay (Fig. 1b).

To extend this, we used an inhibitor against WNK1. WNK463 (WNKi), a selective WNK inhibitor[10], completely blocked CCL21-induced phosphorylation of OXSR1 on S325, a known target of WNK1 activity[11] (Supplementary Fig. 2d). WNKi showed a dose-dependent inhibition of CCL21-induced migration speed and shape polarization, with the extended cellular morphology of migrating T cells becoming more circular at higher WNKi doses (Supplementary Fig. 2e, f, Fig. 1c). 5 μM WNKi resulted in a maximal inhibition of CCL21-induced migration and polarization, 0.5 μM WNKi partially inhibited migration and polarization, whereas 0.2 μM WNK1 reduced migration speed by a small amount but had no effect on polarization.

To analyze the role of the SLC12A-family of ion co-transporters, which are direct phosphorylation targets of OXSR1 and STK39 and at least five of which are expressed in CD4[+] T cells (Supplementary Fig. 2g), we made use of both genetic mutation and an inhibitor. SLC12A2-deficient CD4[+] T cells migrated more slowly in response to CCL21 compared to control SLC12A2-expressing cells, albeit the reduction in speed was smaller than that seen in WNK1-deficient cells (Fig. 1d). Furthermore, treatment of T cells with bumetanide, an SLC12A2 inhibitor that blocks ion movement (SLC12A2i), also caused decreased CCL21-induced migration speed (Fig. 1e). Polarization of the cell, however, was not affected (Supplementary Fig. 2h). These results support a role for the SLC12A2 ion co-transporter in T cell migration.

We extended these studies to CCL21-induced migration of CD4[+] T cells in 3D in a collagen-I matrix. Again, naive WNK1-deficient CD4[+] T cells, or T cells treated with WNKi or SLC12A2i showed reduced speeds of migration (Supplementary Movie 2, Supplementary Fig. 3a–c). Lastly, we asked if the requirement for WNK1 in chemokine-induced migration also applied to activated CD4[+] T cells. We found that CXCL12-induced migration of activated CD4[+] T cells under agarose was strongly reduced in the absence of WNK1, or by its inhibition (Supplementary Fig. 2i). Furthermore, in a collagen-I matrix, CXCL12-induced chemotaxis (directional movement towards the chemokine in a chemokine gradient) of activated CD4[+] T cells was reduced by inhibition of WNK1 (Supplementary Fig. 2j). Thus, like naive CD4[+] T cells, activated T cells also require WNK1 for chemokine-induced migration. Taken together, these results show that WNK1, OXSR1, STK39 and ion influx through SLC12A2 are required for CCL21-induced migration of T cells in confinement.

### WNK1-regulated ion movement and its requirement for CD4[+] T cell migration

The above results predict that CCL21 stimulation activates the WNK1 pathway, resulting in Na[+], K[+] and Cl[–] ion entry into the T cell, and that these ions are required for migration. To directly determine if chemokine stimulation induces ion movement in T cells, we used inductively-coupled plasma mass spectrometry (ICP-MS) to measure element concentrations in cell lysates and used these as a proxy for the abundance of ions. This showed that treatment with CCL21 resulted in an increase in K[+] ions (Fig. 1f). Inhibition with WNKi reduced the amount of K[+] in the T cells and blocked the CCL21-induced increase. It was not possible to reliably measure influx of Na[+] or Cl[–] ions into the cells for technical reasons relating to high amounts of these ions in the extracellular medium. Thus, CCL21 stimulation activates a WNK1-dependent pathway that results in K[+] entry into T cells, presumably in part through SLC12A2 or other SLC12A-family ion co-transporters.

To directly investigate if ions are required for T cell migration, we reduced the concentration of Na[+] and Cl[–] ions in the medium while maintaining isotonicity. CCL21-induced migration of CD4[+] T cells was substantially reduced as the concentrations of Na[+] and Cl[–] ions were

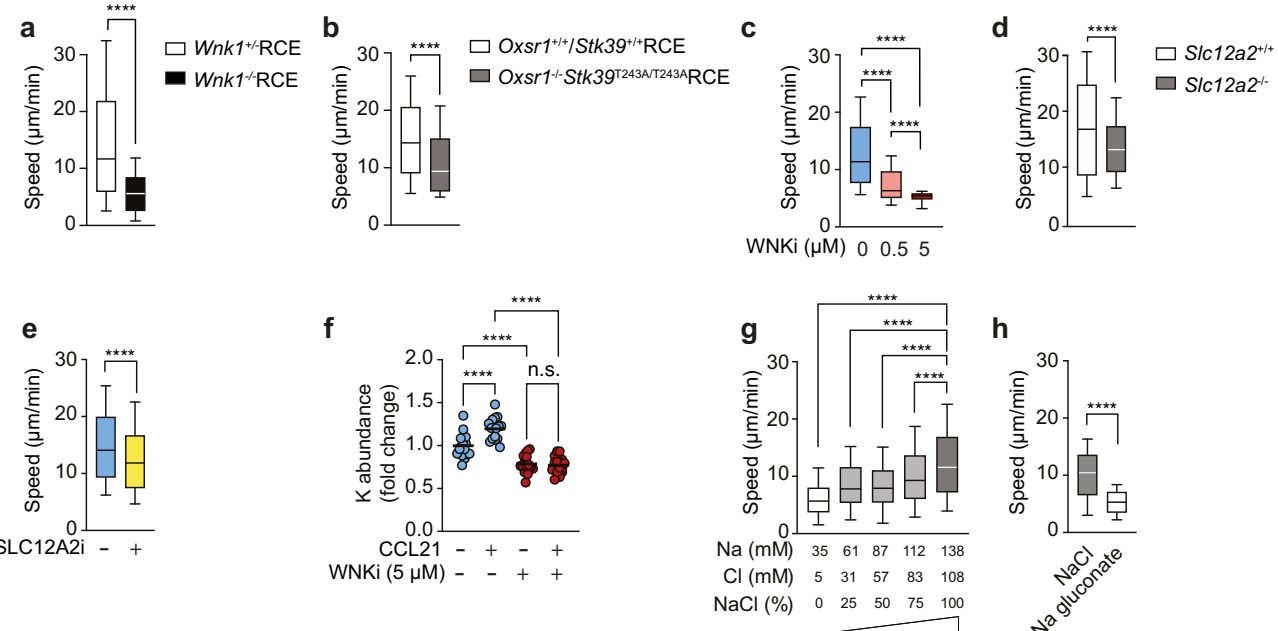

**Fig. 1 | The WNK1 pathway is required for CCL21-induced ion influx-dependent migration. a–e** Speed of mouse naive CD4+ T cells of the indicated genotypes or treated with inhibitors migrating on ICAM-1 under agarose in response to CCL21. **f** Inductively-coupled plasma mass spectrometry measurements of K in lysates of mouse naive CD4+ T cells, either vehicle or WNKi-treated in response to the presence or absence of CCL21 stimulation for 10 min. Each dot represents 1 biological replicate, lines represent mean. Speed of naive CD4+ T cells in media with varying levels of NaCl, with isotonicity maintained by adding D-sorbitol (**g**) or in media containing either NaCl or Na gluconate (**h**). In **g**, the concentrations of Na and Cl comprise NaCl as well other components of the medium. WNKi, WNK inhibitor (WNK463); SLC12A2i, SLC12A2 inhibitor (bumetanide). Sample numbers: *Wnk1*+/−RCE, *n* = 1714 cells; *Wnk1*−/−RCE, *n* = 1757 cells (**a**); *Oxsr1*+/+/*Stk39*+/+RCE, *n* = 1238

cells; *Oxsr1*−/−/*Stk39*T234A/T234ARCE, *n* = 2223 cells (**b**); Uninhibited, *n* = 532 cells; 0.5 μM WNKi, *n* = 752 cells; 5 μM WNKi, *n* = 579 cells (**c**); *Slc12a2*+/+, *n* = 2826 cells; *Slc12a2*−/−, *n* = 1650 cells (**d**); Vehicle, *n* = 2281 cells, SLC12A2i, *n* = 1669 cells (**e**); *n* = 16 mice (**f**); 0% NaCl, *n* = 2975 cells; 25% NaCl, *n* = 2307 cells; 50% NaCl, *n* = 4881 cells; 75% NaCl, *n* = 3715 cells; 100% NaCl, *n* = 5498 cells (**g**); NaCl, *n* = 1755 cells; Na gluconate, *n* = 1494 cells (**h**). Data pooled from at least 2 (**a**, **d**, **e**, **g**) or 4 (**f**) independent experiments, or 1 experiment representative of 2 (**c**, **h**) or 3 (**b**). Box-plot center line, median; box limits, 75th and lower 25th percentiles; whiskers, 10th and 90th percentiles. Statistical analysis (all 2-sided) was carried using Mann-Whitney (**a**, **b**, **d**, **e**, **h**) to generate *p*-values, and Kruskal–Wallis (**c**, **g**) or 1-way ANOVA (**f**) to generate FDR-adjusted *q*-values; n.s. not significant, ****p or q < 0.0001. Source data are provided as a Source Data file.

decreased (Fig. 1g). A similar effect was seen if we replaced Cl− ions with gluconate (Fig. 1h). Taken together, these results show that CCL21 stimulation results in WNK1-dependent ion entry into T cells, and that Na+ and Cl− ions are required for migration.

## WNK1 pathway proteins regulate water entry and cell volume
WNK1-induced ion influx would be predicted to cause water entry by osmosis. In support of this, WNK1 has been shown to regulate the volume of non-lymphoid cells, allowing rapid responses to changing extracellular osmolarity by controlling movement of ions and hence water across the plasma membrane[12]. Thus, we hypothesized that in response to CCL21, WNK1-mediated ion influx would cause water entry, which may be required for migration. To determine if the WNK1 pathway regulates water movement in CD4+ T cells, we first measured the relative volume of cells with defects in the pathway using a Coulter principle-based cell counter. CCL21 induced a volume increase in WNK1-expressing T cells, whereas no such increase was seen in the absence of WNK1, or if WNK1 was inhibited (Fig. 2a, b). Furthermore, WNK1-deficient or WNK1-inhibited T cells were about 15% smaller even in the absence of CCL21, indicating a basal homeostatic function for WNK1 in regulating cell volume. It is likely that WNK1 is required for regulatory volume increase, a mechanism by which cells recover in volume after exposure to a hypertonic environment[4]. Mutation of *Oxsr1* and *Stk39* also caused T cells to become smaller and to no longer respond to CCL21 by increasing in volume (Fig. 2c). In contrast, SLC12A2i did not change basal T cell volume, but blocked the CCL21-induced volume increase (Fig. 2d), consistent with this inhibitor having the smallest effect on migration speed (Fig. 1e). To extend this analysis, we measured cell volume by imaging T cells migrating in response to

CCL21. This showed that CCL21 induced a volume increase which was blocked by treatment with WNKi, and that the inhibitor also caused a decrease in basal T cell volume, consistent with the Coulter counter data, and in agreement with a role for WNK1 in homeostatic volume regulation (Fig. 2e, f).

To directly evaluate if chemokine stimulation results in water entry into CD4+ T cells, we used NMR to measure uptake of 2H2O into the cells. Similar to the volume analysis, CCL21 stimulation resulted in an increase in 2H2O in the cell, which was blocked by treatment with WNKi (Fig. 2g). Taken together, these results demonstrate that WNK1, OXSR1, STK39 and SLC12A2 regulate the volume of T cells in response to CCL21 stimulation, and that WNK1 regulates CCL21-induced water entry.

## AQP3 is required for CCL21-induced T cell migration
Water enters cells through aquaporin (AQP) channels, a family of 11 proteins in the mouse. RNA-seq analysis shows that naive CD4+ T cells predominantly express *Aqp3* and lower levels of *Aqp9* and *Aqp11* (Fig. 2h). To investigate the importance of water movement for T cell migration we used DFP00173 (AQP3i), a selective AQP3 inhibitor[13]. DFP00173 inhibits AQP3, but not AQP7 or AQP9; it is not known if it inhibits AQP11. Notably, AQP3i treatment caused a significant decrease in the speed of CCL21-induced migration of T cells under agarose or in a collagen-I matrix, in agreement with previous studies of AQP3-deficient T cells[14] (Fig. 2i, Supplementary Fig. 3c). Treatment with AQP3i also reduced CCL21-induced polarization as measured by an increase in circularity (Supplementary Fig. 2h). Measurement of cell volume showed that like SLC12A2i, AQP3i did not change the basal volume of T cells but eliminated the CCL21-

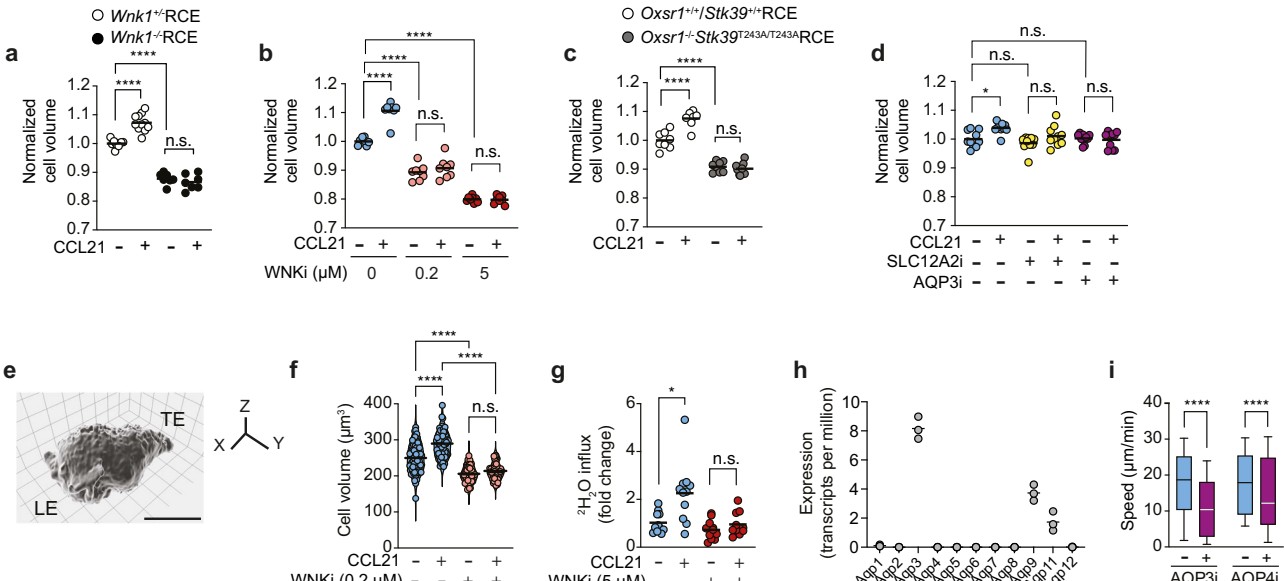

**Fig. 2 | The WNK1 pathway controls chemokine-dependent water influx which is required for T cell migration.** a–d Normalized cell volume of naive CD4[+] T cells of the indicated genotypes, or treated with inhibitors, stimulated with CCL21 for 20 min where indicated. Each dot represents the modal cell volume of ±2000 cells determined using a Casyton, lines represent means. **e** Representative 3D reconstructed image from deconvolved instant structured illumination microscopy (iSIM) data of a mouse naïve CD4[+] T cell expressing tdTomato-CAAX to visualize the plasma membrane, migrating on ICAM-1 in response to CCL21. Scale bar, 5 μm. LE leading edge; TE trailing edge. **f** Cell volume quantification of 3D reconstructions of iSIM images of the cells such as the one shown in **e**, migrating in response to CCL21, with or without WNKi. Each dot represents 1 cell, lines represent mean. **g** Fold change in [2]H$_2$O levels in lysates of naive CD4[+] T cells stimulated (or not) with CCL21 in [2]H$_2$O-based medium for 10 min, with or without WNKi. Fold change expressed relative to mean level of [2]H$_2$O in T cells without CCL21 stimulation and in the absence of WNKi. Each dot represents 1 biological replicate, lines represent mean.

**h** mRNA levels of *Aqp* genes in mouse naïve CD4[+] T cells from RNA-seq analysis[6]. Lines represent mean. **i** Speed of naive CD4[+] T cells treated with either AQP3i or AQP4i and their respective vehicle controls. WNKi, WNK inhibitor (WNK463); SLC12A2i, SLC12A2 inhibitor (bumetanide); AQP3i, AQP3 inhibitor (DFP00173); AQP4i, AQP4 inhibitor (AER-270). Sample numbers: *Wnk1*[+/−]RCE, *n* = 10 mice, *Wnk*[−/−]RCE, *n* = 7 mice (**a**); *n* = 8 mice/condition/genotype (**b**, **c**); *n* = 8 mice/condition (**d**); Vehicle unstimulated, *n* = 71 cells, Vehicle CCL21, *n* = 69 cells, WNKi unstimulated, *n* = 42 cells; WNKi CCL21, *n* = 52 cells (**f**); 11 biological replicates, each consisting of cells pooled from 2 mice (**g**); 3 mice (**h**); AQP3i (vehicle only), *n* = 573 cells, AQP3i, *n* = 611 cells, AQP4i (vehicle only), *n* = 657 cells, AQP4i, *n* = 1527 cells (**i**). Data pooled from at least 3 (**a**–**d**) or 2 (**f**, **g**, **i**) experiments. Box-plot center line, median; box limits, 75th and lower 25th percentiles; whiskers, 10th and 90th percentiles. Statistical analysis was carried using a Kruskal–Wallis test (**a**–**d**, **f**, **g**, **i**) to generate FDR-adjusted *q*-values; n.s. not significant, *0.01 <*q* < 0.05, ****q* < 0.0001. Source data are provided as a Source Data file.

induced volume increase (Fig. 2d). A previous study had also proposed a role for AQP4 in T cell trafficking[15]. Using AER-270 (AQP4i) an AQP4 inhibitor[16], we were able to show that treatment with this compound also reduced CCL21-induced migration speed, despite a lack of *Aqp4* expression in the RNA-seq data (Fig. 2h, i). We conclude that either AQP4 protein is indeed expressed despite undetectable levels of *Aqp4* mRNA, or that the AQP4i has off-target effects, perhaps on other AQPs.

AQP3 is a channel for H$_2$O$_2$ as well as water[17]. A previous study suggested that AQP3 is required for T cell migration because it facilitates H$_2$O$_2$ influx[14]. To investigate if H$_2$O$_2$ is required for mouse naive CD4[+] T cell migration, we treated T cells with 20 U/ml catalase, a concentration sufficient to eliminate exogenously added 1 mM H$_2$O$_2$ (Supplementary Fig. 4a). Treatment of T cells with this concentration of catalase did not impair migration in response to CCL21 (Supplementary Fig. 4b), implying that H$_2$O$_2$ is not required for T cell migration. It is not clear why our results differ from the earlier publication. However, we note several methodological differences. Whereas we directly measured migration speed in response to CCL21 under agarose, the earlier publication measured % migration through a Transwell in response to CXCL12. Furthermore, cells were incubated with 2000 U/ml catalase, 100-fold more than the concentration we used, which may have off-target effects. Taken together, these results show that AQP3 is required for CCL21-induced migration in CD4[+] T cells, likely as a result of water transport, supporting the hypothesis that WNK1-regulated water entry is required for chemokine-induced T cell migration.

## WNK1 pathway proteins polarize to the leading edge of migrating CD4[+] T cells

To further understand how WNK pathway proteins may regulate chemokine-induced migration, we evaluated the sub-cellular distribution of the proteins and their activities in CD4[+] T cells migrating in confinement under agarose. Live cell imaging of T cells expressing GFP-tagged WNK1, OXSR1 and SLC12A2 migrating in response to CCL21 showed that all three proteins were more abundant at the leading edge of the cell compared to the distribution of GFP only (Supplementary Movie 3, Fig. 3a–c). This accumulation at the leading edge was not a function of nuclear exclusion or passive enrichment at the leading edge of the fusion proteins, since nuclear-excluded GFP was enriched in the trailing part of the cell (Fig. 3d–f). Furthermore, imaging of fixed cells demonstrated that in migrating T cells, OXSR1, SLC12A2 and AQP3 are enriched at the leading edge and depleted at the trailing edge as identified by localization of CDC42 and CD44, respectively (Fig. 3g, h, Supplementary Fig. 5a, b). To analyze activity of the pathway we imaged the distribution of phosphorylated WNK1 (S382), OXSR1 (S325), and SLC12A2 (T203/T207/T212), phosphorylations that indicate increased activity of these proteins[11, 18, 19]. Notably, these phospho-proteins were also enriched at the leading edge of migrating T cells (Fig. 3g, h, Supplementary Fig. 5a, b). The polarization of these WNK1 pathway proteins and their activities was eliminated in WNK1-deficient or WNK1-inhibited cells (Supplementary Fig. 6a–j). Thus, in T cells migrating in response to CCL21, WNK1 pathway proteins and their activities accumulate at the leading edge of the cell, implying that CCR7 signaling via WNK1, OXSR1 and STK39 is likely to

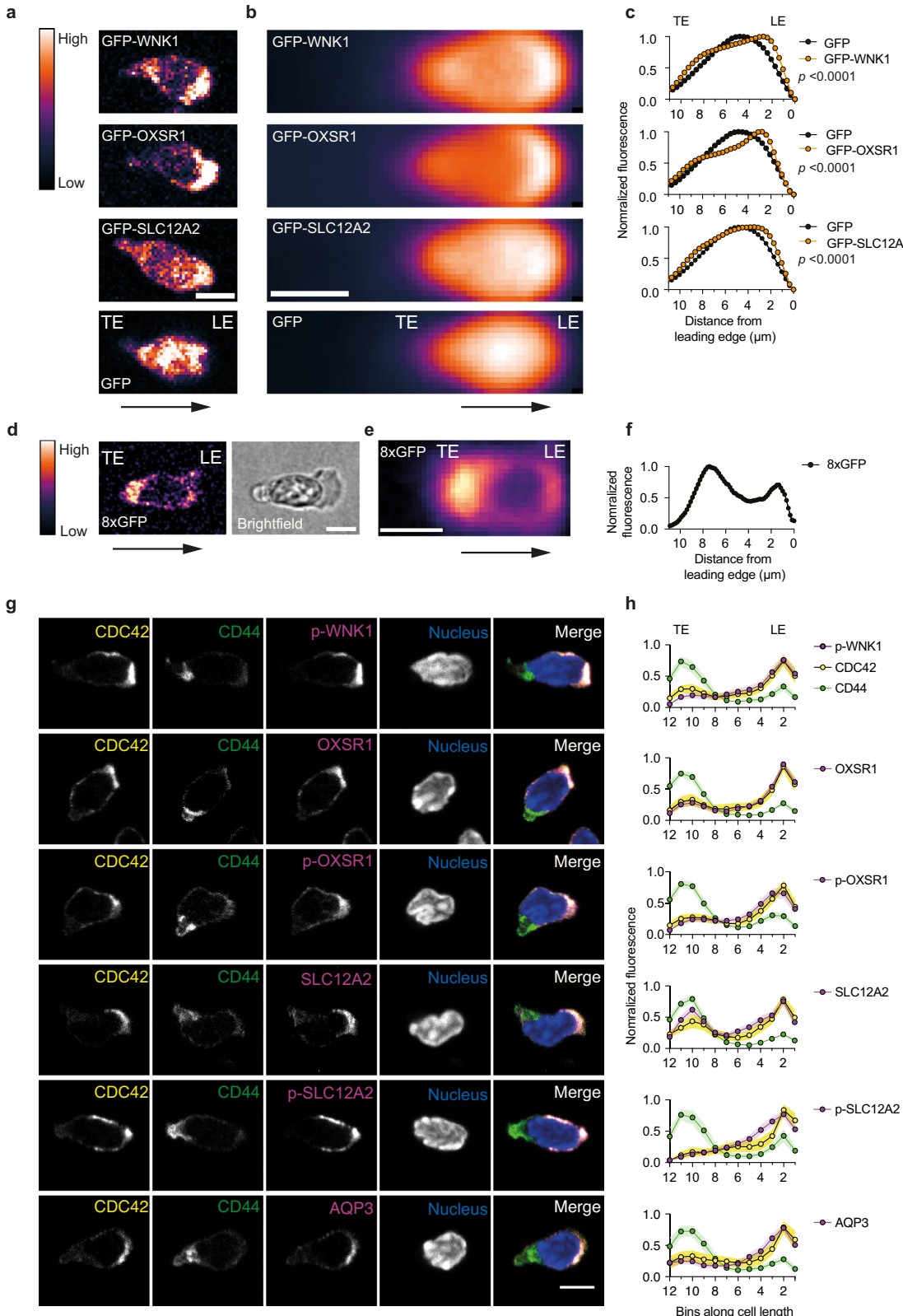

lead to localized entry of Na$^+$, K$^+$ and Cl$^-$ ions via SLC12A2 at the leading edge of the migrating cells.

## The WNK1 pathway regulates spacing between the plasma membrane and the F-actin cytoskeleton

Migration of T cells requires actin polymerization at the leading edge which results in F-actin filaments pushing forward against the plasma membrane[3,20]. To determine if this process is controlled by the WNK1 pathway, we measured total F-actin in CCL21-stimulated CD4$^+$ T cells using flow cytometry. This showed a rapid CCL21-induced increase in F-actin which was partially decreased by inhibition of WNK1 and SLC12A2 and by *Oxsr1* and *Stk39* mutations (Supplementary Fig. 7a–c), implying that the WNK1 pathway and ion influx may regulate actin polymerization.

**Fig. 3 | WNK1 pathway proteins polarize to the leading edge of migrating CD4⁺ T cells. a–f** Naive CD4⁺ T cells constitutively expressing GFP-tagged WNK1 pathway proteins or GFP alone (**a–c**) or a nucleus-excluded concatemer of 8 GFP molecules (8xGFP) (**d–f**) were imaged migrating on ICAM-1 under agarose in response to CCL21; example images are shown (**a**, **d**). Arrow indicates direction of migration. A brightfield image is included in **d** to show the location of the nucleus. Individual cell images were aligned in the direction of migration and average intensity-projected to visualize GFP distribution (**b**, **e**). Intensity quantification of GFP-tagged proteins along the full width of the cell compared to EV-GFP (**c**) or of 8xGFP along the full width of the cell (**f**), with the lines in the graphs representing mean (±95% CI) normalized fluorescence over many cells. Note that the 95% CI is so small that it is not visible around the mean line. *p*-values for difference in distribution of GFP-tagged WNK1, OXSR1 and SLC12A2 compared to GFP alone were determined using a 2-way ANOVA. **g, h** Naive CD4⁺ T cells migrating in response to CCL21 on ICAM-1 were fixed and stained with antibodies against the indicated (phospho−) proteins. Typical example images of 6 cells showing staining for individual proteins and a merged image (**g**). Mean (±95% CI) normalized fluorescence intensity over many cells of WNK1 pathway proteins in relation to the LE and TE, as indicated by highest levels of CDC42 and CD44 respectively (**h**). Cell lengths were binned into 12 bins per cell. Scale bars, 5 µm. LE leading edge; TE, trailing edge. Sample numbers: GFP-WNK1, *n* = 1244 cell shapes; GFP-OXSR1, *n* = 2557 cell shapes; GFP-SLC12A2, *n* = 2564 cell shapes; GFP, *n* = 2345 cell shapes (**c**); 8xGFP, *n* = 242 cells (**f**); p-WNK1, *n* = 47 cells; OXSR1, *n* = 68 cells; p-OXSR1, *n* = 59 cells; SLC12A2, *n* = 58 cells; p-SLC12A2, *n* = 62 cells; AQP3, *n* = 62 cells (**h**). Data are from 1 experiment, representative of at least 2 experiments (**b**, **c**, **f**); pooled from 2 (OXSR1, SLC12A2) or 3 (p-WNK1, p-OXSR1, p-SLC12A2, AQP3) experiments. Source data are provided as a Source Data file.

Forward protrusion of actin filaments at the leading edge is generated by G-actin monomer addition to F-actin filament tips, most likely through a Brownian ratchet mechanism[21–23]. Thermal (Brownian) motion of both the membrane and the filaments results in a fluctuating gap, and when this exceeds 2.7 nm, an actin monomer can be added to the filament tip, thereby extending it. This prevents the membrane from returning to its original position, resulting in incremental forward protrusion. We hypothesized that WNK1-induced water entry at the leading edge would result in localized swelling of the membrane, pulling it away from underlying F-actin, thereby creating space into which actin filaments can polymerize, a mechanism we term a facilitated Brownian ratchet. To directly test if the WNK1-induced water movement affects spacing between the plasma membrane and the underlying F-actin, we first used instant structured illumination microscopy, a super-resolution method, to image the plasma membrane and F-actin in CD4⁺ T cells migrating in response to CCL21[24]. We found that at the leading edge the distance from the plasma membrane to the peak F-actin signal was around 174 nm, whereas at the side of the cell, this distance was around 43 nm, consistent with an increased spacing at the leading edge (Fig. 4a, b), although we note that this increased spacing could also be caused by a wider band of F-actin at the leading edge. Notably, treatment of the T cells with SLC12A2i or AQP3i caused a significant decrease in the membrane to peak F-actin spacing, with AQP3i again having the stronger effect (Supplementary Movie 4, Fig. 4c, d). Importantly, we noted a significant correlation between membrane to peak F-actin spacing and cell speed, supporting our hypothesis that this spacing is critical for T cell migration (Fig. 4e). Thus, our data suggest that SLC12A2 and AQP3 function are required to maintain a larger membrane to peak F-actin spacing at the leading edge and are consistent with water entry causing increased spacing between the membrane and the underlying F-actin. However, we cannot rule out that the changes in these distances are caused by an altered width of cortical F-actin in the migrating cells which could also correlate with migration speed.

## WNK1 activity and ion and water influx regulate membrane proximal F-actin at the leading edge

Surprisingly, analysis of F-actin in migrating cells has shown that despite greatly elevated levels of F-actin at the front of the cell, there is a lower density of membrane-proximal actin (MPA) within 10 nm of the leading edge compared to other parts of the migrating cell[25]. This is possibly due to a reduced load of the membrane on actin filaments at the leading edge, which results in rapid polymerization of actin filaments perpendicular to the membrane. These outcompete a network of filaments at shallow angles, thereby generating a region with lower F-actin density proximal to the plasma membrane[26]. We hypothesized that WNK1-induced water entry and membrane swelling may contribute to the reduced load and hence lower MPA at the leading edge. To test this, we analyzed migrating T cells expressing two fluorescent probes: MPAct-mCherry (MPAct) and GFP-CaaX (CaaX)[25]. Both contain fluorescent proteins that are tethered to the plasma membrane, but MPAct also contains an F-actin-binding domain. Thus, CaaX is evenly distributed on the plasma membrane, whereas MPAct accumulates in regions of the membrane with higher concentrations of underlying MPA that is within 10 nm of the membrane. Hence the MPAct/CaaX ratio is a measure of the distribution of MPA. Analysis of T cells expressing MPAct and CaaX migrating in response to CCL21, showed that, as expected, the MPAct/CaaX ratio is reduced at the leading edge compared to the side and trailing edges (Fig. 5a, b). Notably, treatment of T cells with WNKi, SLC12A2i or AQP3i resulted in a significant increase in MPA at the leading edge, with less polarization of the MPA signal between the front and back of the cell (Supplementary Movie 5, Supplementary Fig. 7d, e, Fig. 5c–f). Importantly, for these studies we used a low dose of WNKi (0.2 µM) which reduces migration speed but still allows cells to polarize (Supplementary Fig. 2e). Thus, the activities of WNK1, SLC12A2 and AQP3 are required to maintain reduced MPA at the leading edge of T cells, consistent with our hypothesis that localized ion influx and subsequent water entry causes the membrane to swell away from the underlying F-actin.

## WNK1 pathway regulates actin retrograde flow in migrating T cells

If WNK1-induced water entry at the leading edge is required to generate space and hence facilitate the Brownian ratchet mechanism of actin filament growth, inhibition of the pathway should affect the rate of actin polymerization at the membrane. Continuous addition of G-actin monomers at the leading edge of migrating cells results in retrograde flow of F-actin, the rate of which can thus be used as a surrogate for actin polymerization rate at the front of the cell[27]. To determine if the WNK1 pathway regulates this process, we measured rates of retrograde actin flow in CCL21-stimulated T cells expressing LifeAct-eGFP to visualize F-actin. The cells were imaged responding to CCL21 on PEG-coated glass, an arrangement that eliminates integrin-based adhesion, causing the cell to stay in place and the F-actin cytoskeleton to slip in relation to the substrate. In contrast to a partial decrease in total F-actin, retrograde actin flow was strongly decreased in WNK1-deficient or WNK1-inhibited T cells (Supplementary Movie 6, Supplementary Fig. 8a, b, Fig. 6a, b). Furthermore, both SLC12A2i and AQP3i partially slowed retrograde actin flow, with AQP3i having the stronger effect. Thus, WNK1, SLC12A2 and AQP3 function are required for actin retrograde flow, consistent with our hypothesis that ion and water influx influence the rate of addition of G-actin monomers to the growing tips of F-actin filaments at the leading edge.

## WNK1-induced water entry is required for T cell migration

To directly determine if WNK1-induced water entry is required for migration, we asked if forcing water into the cells by reducing the tonicity of the medium could rescue migration in WNK1-inhibited cells. This volume increase in response to hypotonic medium would counteract the defective regulatory volume increase observed in the

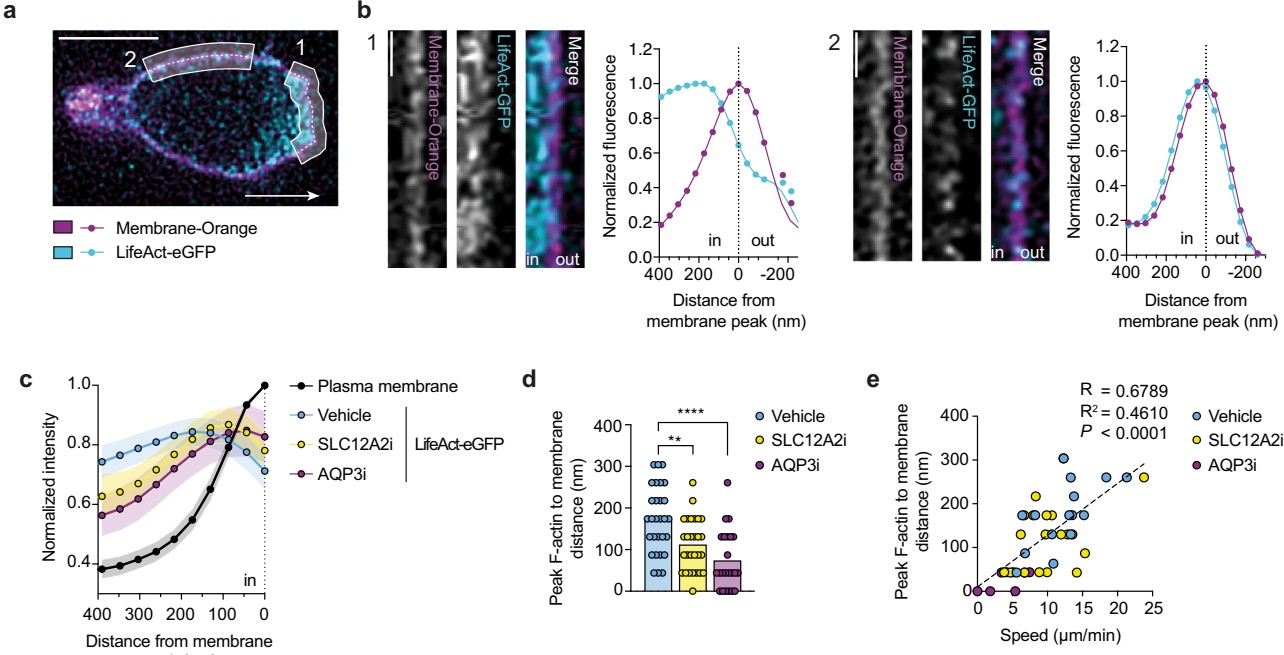

**Fig. 4 | The WNK1 pathway regulates spacing between the plasma membrane and the F-actin cytoskeleton. a-e** LifeAct-eGFP expressing naïve CD4[+] T cells stained with CellMask Orange to show plasma membrane (PM), migrating on ICAM-1 coated dishes under CCL21-containing agarose were imaged by iSIM. Representative deconvoluted image of a migrating cell indicating a region of the leading edge (1) and the side of the cell (2); white arrow shows direction of migration; scale bar, 5 μm (**a**). Fluorescence images of linearized regions 1 and 2 from **a** and graphs showing normalized LifeAct-eGFP and PM fluorescence as a function of distance from peak PM fluorescence, indicated by the dotted line, with normalization setting the peak fluorescence to 1; scale bar, 1 μm (**b**). Mean (±95% CI) LifeAct-eGFP and PM fluorescence as a function of distance from peak PM fluorescence (dotted line) in CD4[+] T cells treated with inhibitors or vehicle (**c**). The mean peak normalized

LifeAct-eGFP fluorescence is less than 1 because the signal is averaged over many cells and the position of the peak varies from cell to cell. Distance of peak LifeAct-eGFP fluorescence to the PM; each dot represents a single frame from one cell, columns represent mean (**d**). Distance of peak LifeAct-eGFP fluorescence to the PM as a function of migration speed in CD4[+] T cells treated with inhibitors or vehicle; dashed line, linear regression; R, correlation coefficient (**e**). SLC12A2i, SLC12A2 inhibitor (bumetanide); AQP3i, AQP3 inhibitor (DFP00173). Sample numbers: Vehicle, $n = 29$ cells; SLC12A2i, $n = 28$ cells; AQP3i, $n = 23$ cells (**c**, **d**); Vehicle, $n = 16$ cells; SLC12A2i, $n = 20$ cells; AQP3i, $n = 6$ cells (**e**). Data are pooled from 3 (AQP3i) or 4 (Vehicle, SLC12A2i) experiments. Statistical analysis carried out using 1-way ANOVA (**d**) or Pearson's correlation test (**e**), **0.001 < $q$ < 0.01, ****$q$ < 0.0001. Source data are provided as a Source Data file.

absence of WNK1. For these studies we again used a low dose of WNKi to slow the cells down, but still allow them to polarize. Remarkably, while reducing tonicity caused a reduction of migration speed in T cells treated with vehicle control, it increased migration speed in cells where WNK1 had been partially inhibited. This was most effective at 248 mOsm/l, a ~15% reduction of tonicity compared to isotonic medium (292 mOsm/l) (Fig. 6c). The hypotonic rescue of migration in WNK1-inihibited T cells could be due to the increased water entry, or, alternatively, a result of reduced NaCl concentration. For example, since Cl[−] is an inhibitor of WNK1[12, 28], it is possible that the reduced Cl[−] concentration in the hypotonic medium could lead to reduced intracellular Cl[−] thereby increasing WNK1 activity and overcoming the effect of a low dose (0.2 μM) of WNKi. To distinguish if the rescue of migration was caused by increased water or entry or decreased NaCl, we compared the migration speed of WNK1-inhibited T cells in hypotonic medium at 248 mOsm/l or in isotonic medium with the same reduced amount of NaCl but with isotonic osmolarity restored using L-glucose, an inert osmolyte (Fig. 6d). Whereas hypotonic medium rescued most of the migration defect of WNK1-inhibited cells, isotonic medium with reduced NaCl was less effective, demonstrating that water entry is required for T cell migration. One caveat to this conclusion is that, depending on the chloride conductance of the cell, the isotonic medium with reduced NaCl may result in less reduction of intracellular chloride compared to the hypotonic medium.

In further experiments, we found that treatment of WNK1-inhibited cells with hypotonic medium also restored lower MPA levels at the leading edge of migrating cells (Fig. 5c, e). Finally, the reduced retrograde actin flow in WNK1-inhibited T cells was partially

rescued when the cells were placed in hypotonic medium (Fig. 6e). Taken together, these results support our hypothesis that WNK1-induced osmotic water entry is required for CCL21-induced actin retrograde flow and T cell migration.

## Discussion

We show that chemokine receptor signaling in migrating T cells activates WNK1, transducing signals via OXSR1 and STK39 and the SLC12A2 ion co-transporter which results in ion influx, and subsequent water entry by osmosis, most likely via aquaporins. Importantly, we have been able to directly demonstrate chemokine-induced ion and water entry into the cell and show that these are required for cell migration. Furthermore, we show that the WNK1 pathway is activated at the leading edge of migrating T cells and that since AQP3 is localized to the front of the cell, water may also enter at the leading edge. We show that WNK1 pathway function is required for the reduction in membrane-proximal actin at the leading edge that correlates with cell migration. We hypothesize that WNK1-regulated ion and water influx at the leading edge is required to cause swelling of the plasma membrane, allowing it to move away from the underlying F-actin, thereby generating space into which actin filaments can extend and facilitating a Brownian ratchet mechanism that is essential for actin polymerization and hence forward cell movement (Fig. 6f).

We note that loss or inhibition of WNK1 results in a stronger defect in migration than mutations in OXSR1 and STK39, or inhibition of SLC12A2 or AQP3. This may be due to redundancy, with other signaling proteins taking the place of OXSR1 and STK39 or to other SLC12A- or AQP-family members compensating for inhibition of SLC12A2 and

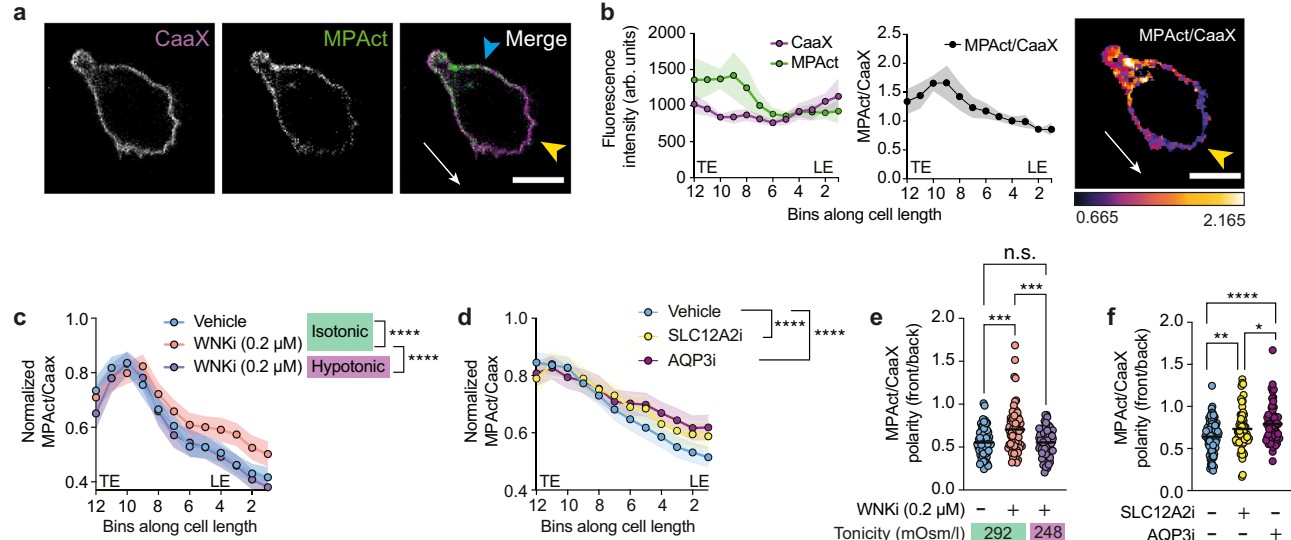

**Fig. 5 | WNK1-mediated water influx reduces membrane proximal F-actin at the leading edge. a–f** Naive CD4[+] T cells co-expressing GFP-CaaX (CaaX) and MPAct-mCherry (MPAct), were imaged migrating in response to CCL21 on ICAM-1 under agarose. Representative images of CaaX and MPAct fluorescence in a cell showing direction of migration (white arrow), LE (yellow arrowhead) and side of cell (blue arrowhead) (**a**). Mean (±95% CI) CaaX and MPAct fluorescence and MPAct/CaaX ratio in the same cell as shown in **a**, averaged over 20 frames taken every 15 s displayed as a function of length along cell; image shows MPAct/CAAX ratio of the cell in **a** (**b**). arb. units, arbitrary units. Mean (±95% CI) normalized MPAct/CaaX ratios along full cell length and width taken from multiple cells treated with inhibitor or vehicle at the indicated tonicities of medium (**c, d**). MPAct/CaaX polarity ratio, where the mean of the front 3 bins was divided by the mean of the back 3 bins using the same data as in **c, d**; each dot shows data from one cell, line shows mean (**e, f**). WNKi, WNK inhibitor (WNK463); SLC12A2i, SLC12A2 inhibitor (bumetanide); AQP3i, AQP3 inhibitor (DFP00173). LE leading edge, TE trailing edge. Scale bars, 5 μm. Sample numbers: Vehicle isotonic, *n* = 80 cells; WNKi isotonic, *n* = 79 cells; WNKi hypotonic, *n* = 46 cells (**c**); Vehicle, *n* = 88 cells; SLC12A2i, *n* = 63 cells; AQP3i, 61 *n* = cells (**d, f**); Vehicle isotonic, *n* = 71 cells; WNKi isotonic, *n* = 71 cells; WNKi hyptonic, *n* = 65 cells (**e**). Data are pooled from 3 (**c, d** AQP3i, **e**, and **f** AQP3i) or 4 (**d, f** Vehicle and SLC12A2i) experiments. 2-way ANOVA (**c, d**), 1-way ANOVA (**e, f**); *0.01 <*q* < 0.05, **0.001 <*q* < 0.01, ***0.0001 <*q* < 0.001, ****q* < 0.0001. Source data are provided as a Source Data file.

AQP3. Alternatively, WNK1 may additionally regulate migration through pathways other than OXSR1, STK39, SLC12A2 and AQP3[29]. Indeed, a high dose of WNK inhibitor or loss of WNK1 results in cells that do not polarize and do not migrate, and whose migration cannot be rescued with hypotonic medium. This implies that WNK1 most likely has an additional function beyond regulating ion and water influx at the leading edge.

The importance of membrane protrusion for forward actin polymerization at the leading edge is supported by studies showing that increased load on lamellipodial F-actin results in a dense actin network with many filaments growing at shallow angles to the membrane[26,30]. In contrast, decreased load results in increased polymerization of actin filaments perpendicular to the membrane, causing rapid forward protrusion of the leading edge. Our studies suggest a mechanism by which such reduced force may be achieved. We demonstrate that CCR7 signaling at the leading edge of T cells results in localized activation of the WNK1 pathway, which we propose would cause water entry and membrane swelling. This would be predicted to decrease the force on the underlying F-actin network and thereby facilitate forward growth of actin filaments.

For this water-facilitated Brownian ratchet to function, water influx at the leading edge would need to cause a local pressure increase which does not immediately equilibrate across the cell, but stays elevated on temporal and length scales relevant to cell motility. In support of this, mammalian cells have been shown to have the characteristics of a poroelastic solid, resembling a fluid-filled sponge[31,32]. The solid phase of this sponge consists of cytoskeleton and organelles whose spatial organization results in pores through which water, small molecules and proteins diffuse. Measurements suggested that in HeLa cells, the solid phase of the cytoplasm has an effective pore size of around 30 nm, which is small enough to slow water flow on time and length scales of 10–100 s and 10 μm respectively, consistent with the requirements for cell migration[31].

Given the broad expression of WNK1[33], similar pathways may play important roles in the migration of many cell types both within the immune system and beyond it. We previously showed that loss of WNK1 slows down the homing of both CD4[+] and CD8[+] T cells to secondary lymphoid organs and impairs CXCL12-induced migration of CD4[−]CD8[−] thymocytes[6,8]. It will be interesting to extend these studies to activated T cells subsets, such as Th1, Th2, Th17, and to investigate the role of WNK1-regulated migration in T cell immune responses. More recently we found that WNK1 is required for CXCL13-induced migration of B cells and their in vivo migration in lymph nodes[34]. Furthermore, previous studies have shown that inhibition of the sodium-hydrogen exchanger NHE-1 (SLC9A1) or AQP9 decreases fMLP-induced migration of human neutrophils, SLC12A2 inhibition reduces migration of glioma and glioblastoma cells, and knockdown of WNK1 or OXSR1 impairs glioma migration, indicating that ion and water movement may be important for migration in multiple contexts[35–38]. Moreover, inhibition of WNK1 was shown to decrease migration of breast cancer cells in vitro and to reduce tumor burden in vivo[39]. However, the mechanistic basis of these observations is unclear. Our results provide a potential explanation for why ion and water movement may be important for cell migration in multiple cell types, including metastatic tumor cells.

More broadly, in a parallel study we have been able to show that T cell antigen receptor (TCR) stimulation of CD4[+] T cells results in the activation of the WNK1 pathway and that the resulting ion and water influx are required for TCR-induced cell proliferation[40]. These results suggest that multiple signaling receptors may co-opt the WNK1 pathway to regulate ion and water movement, which are required for migration, proliferation and perhaps other fundamental physiological processes.

In summary, we showed that chemokine-induced migration of T cells requires WNK1-dependent ion and water influx, and we propose

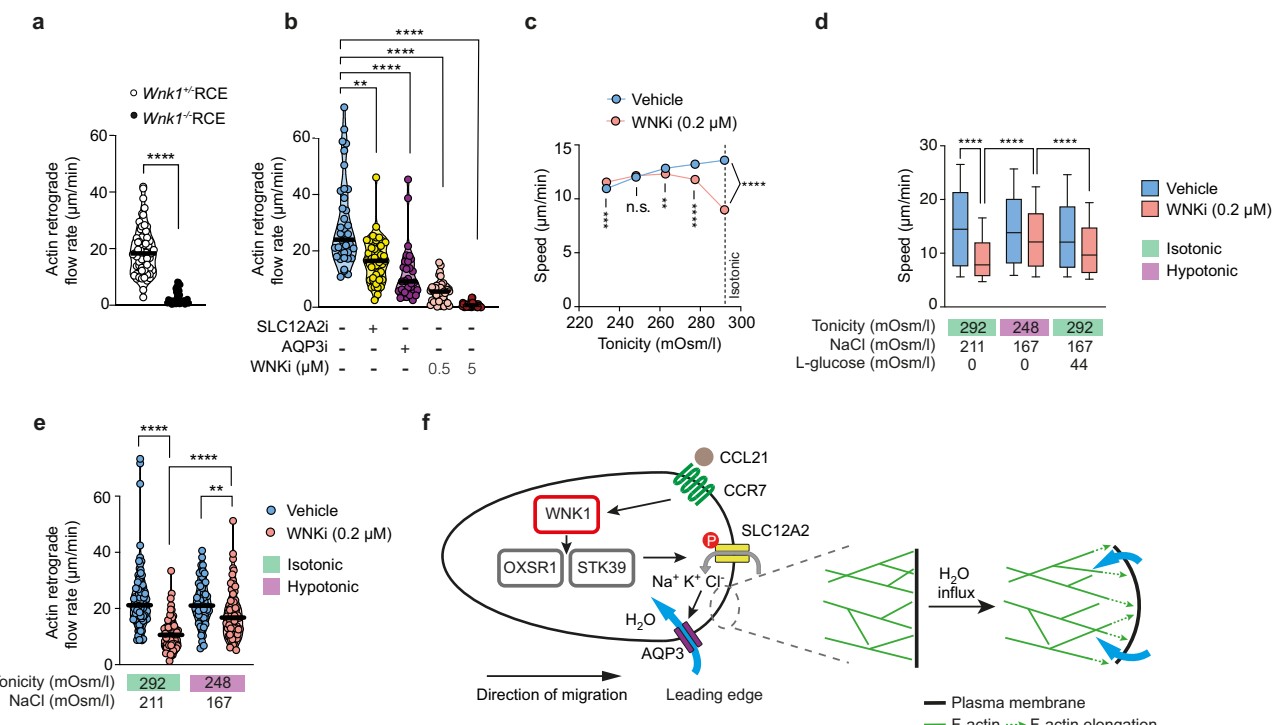

**Fig. 6 | WNK1 pathway-induced water entry is required for retrograde actin flow and T cell migration. a, b** Naive CD4[+] T cells expressing LifeAct-eGFP of the indicated genotypes or treated with inhibitor or vehicle, were stimulated with CCL21 under agarose on a PEG-coated dish and imaged using TIRFM. Dots represent F-actin flow rates within single cells, lines represent median. **c, d** Speed of CD4[+] T cells migrating on ICAM-1 under agarose in response to CCL21 in the presence or absence of WNKi, with tonicity of agarose varied as indicated. Data in **c** is plotted as mean ±95% CI, but the CI is too small to visualize. **e** Retrograde actin flow in CD4[+] T cells migrating in response to CCL21 under agarose on a PEG-coated dish in the presence or absence of WNKi at the indicated tonicities of medium. Dots represent F-actin flow rates within single cells, lines represent median. **f** Proposed model: CCL21 binding to CCR7 at the leading edge of the migrating CD4[+] T cell results in activation of WNK1, which phosphorylates and activates OXSR1 and STK39. These in turn phosphorylate SLC12A-family proteins, resulting in entry of Na[+], K[+] and Cl[−] ions into the cell and consequent water entry through AQP3 (and other aquaporins) by osmosis. Water entry swells the plasma membrane, creating space into which actin filaments can extend, thereby protruding the cell forward. WNKi, WNK inhibitor (WNK463); SLC12A2i, SLC12A2 inhibitor (bumetanide); AQP3i, AQP3 inhibitor

(DFP00173). Sample numbers: $Wnk1^{+/-}$RCE, $n = 61$ cells; $Wnk1^{-/-}$RCE, $n = 31$ cells (**a**); Vehicle, $n = 39$ cells; SLC12A2i, $n = 47$ cells; AQP3i, $n = 31$ cells; WNKi (0.5 μM), $n = 41$ cells; WNKi (5 μM), $n = 37$ cells (**b**); Vehicle 292 mOsm/l, $n = 3077$ cells; Vehicle 277.4 mOsm/l, $n = 2401$ cells; Vehicle 262.8 mOsm/l, $n = 1913$ cells; Vehicle 248.2 mOsm/l, $n = 1803$ cells; Vehicle 233.6 mOsm/l, $n = 2198$ cells; WNKi 292 mOsm/l, $n = 3010$ cells; WNKi 277.4 mOsm/l, $n = 2246$ cells; WNKi 262.8 mOsm/l, $n = 2197$ cells; WNKi 248.2 mOsm/l, $n = 1769$ cells; WNKi 233.6 mOsm/l, $n = 2686$ cells (**c**); Vehicle 292 mOsm/l 211 NaCl, $n = 6168$ cells; WNKi 292 mOsm/l 211 NaCl, $n = 7160$ cells; Vehicle 248 mOsm/l, $n = 6278$ cells; WNKi 248 mOsm/l, $n = 7720$ cells; Vehicle 292 mOsm/l 167 NaCl + L-glucose, $n = 5795$ cells; WNKi 292 mOsm/l 167 NaCl + L-glucose, $n = 7678$ cells (**d**); Vehicle 292 mOsm/l, $n = 83$ cells; WNKi 292 mOsm/l, $n = 72$ cells; Vehicle 248 mOsm/l, $n = 55$ cells; WNKi 248 mOsm/l, $n = 68$ cells (**e**). Data are pooled from at least 2 (**a, b**), 3 (**e**) or 4 (**c, d**) experiments. Box-plot center line, median; box limits, 75th and lower 25th percentiles; whiskers, 10[th] and 90[th] percentiles. 2-sided Mann–Whitney (**a**), Kruskal-Wallis (**b, e**) or 1-way ANOVA (**c, d**); **0.001 < $q$ < 0.01, **** $p$ or $q$ < 0.0001. Source data are provided as a Source Data file.

that water entry at the leading edge of T cells facilitates actin polymerization and hence drives forward cell movement.

## Methods

### Mice

Mice with a conditional allele of *Wnk1* containing loxP sites flanking exon 2 (*Wnk1*[tm1Clhu], *Wnk1*[fl])[7], with a deletion of exon 2 of *Wnk1* (*Wnk1*[tm1.1Clhu], *Wnk1*[−])[6], with a conditional allele of *Oxsr1* containing loxP sites flanking exons 9 and 10 (*Oxsr1*[tm1.1Ssy], *Oxsr1*[fl])[41], with a tamoxifen-inducible Cre in the ROSA26 locus (*Gt(ROSA)26Sor*[tm1(Cre/ESR1)Thl], *ROSA26*[CreERT2], RCE)[42], with an allele of *Stk39* encoding STK39-T243A (*Stk39*[tm1.1Arte], *Stk39*[T243A])[9] that cannot be activated by WNK1, expressing LifeAct-eGFP (Tg(CAG-EGFP)#Rows)[43], expressing a tdTomato protein from the ROSA26 locus (*Gt(ROSA)26Sor*[tm4(ACTB-tdTomato-EGFP)Luo], *ROSA26*-mTmG)[44] or deficient for SLC12A2 (*Slc12a2*[tm1Ges], *Slc12a*[−])[45] or RAG1 (*Rag1*[tm1Mom], *Rag1*[−])[46] have been described before. C57BL/6J mice were provided by the Biological Research Facility of the Francis Crick Institute. All mice were maintained on a C57BL/6J background and intercrossed as required to generate strains with multiple genetically altered alleles. Mice

were kept in rooms with controlled 12 h light/dark cycles at 23 °C and 50% humidity. Mice were housed in individually ventilated cages under specific pathogen-free conditions, and given food and water *ad libitum*. Both female and male mice were used and were at least 6 weeks old at the time of the experiment. All mice were age- and sex-matched within an experiment. All experiments were approved by the Francis Crick Institute Animal Welfare Ethical Review Board and were carried out under the authority of a Project Licence granted by the UK Home Office.

### Bone marrow chimeras

Bone marrow cells from *Wnk1*[fl/+]RCE and *Wnk1*[fl/−]RCE mice or from LifeAct-eGFP*Wnk1*[fl/+]RCE and LifeAct-eGFP*Wnk1*[fl/−]RCE mice or from *Oxsr1*[+/+]*Stk39*[+/+]RCE, *Oxsr1*[fl/fl]*Stk39*[+/+]RCE, and *Oxsr1*[fl/fl]/*Stk39*[T234A/T234A]RCE mice were transferred intravenously into irradiated (5 Gy, [37]Cs source) RAG1-deficient recipient mice which were maintained on 0.02% enrofloxacin (Baytril, Bayer Healthcare) for 4 weeks. In all cases the sex of the donor and recipient mice was matched. The hematopoietic compartment was allowed to reconstitute for at least 8 weeks after bone marrow transfer. In all experiments, control and

experimental chimeric mice or cells derived from them were always sex-matched.

## In vivo deletion of floxed alleles

Tamoxifen treatment of $Wnk1^{fl/+}$RCE and $Wnk1^{fl/-}$RCE bone marrow chimeras was used to generate $Wnk1^{+/-}$RCE and $Wnk1^{-/-}$RCE T cells. Tamoxifen treatment of $Oxsr1^{+/+}Stk39^{+/+}$RCE, $Oxsr1^{fl/fl}Stk39^{+/+}$RCE, and $Oxsr1^{fl/fl}/Stk39^{T234A/T234A}$RCE bone marrow chimeras was used to generate $Oxsr1^{+/+}Stk39^{+/+}$RCE, $Oxsr1^{-/-}Stk39^{+/+}$RCE, and $Oxsr1^{-/-}/Stk39^{T234A/T234A}$RCE T cells, respectively. Mice were injected intraperitoneally with 2 mg tamoxifen in corn oil (Sigma-Aldrich) on 3 ($Wnk1^{fl}$) or 5 ($Oxsr1^{fl}$) consecutive days. Mice were sacrificed 7 ($Wnk1^{fl}$) or 21 ($Oxsr1^{fl}$) days after initial tamoxifen injection to harvest T cells.

## T cell purification and culture

To isolate naïve CD4$^+$ T cells, mouse lymph node cells were incubated with biotin-conjugated antibodies to CD44 (IM7) and CD25 (PC61.5) (both from ThermoFisher, both at 1:400 dilution), CD11c (N418), Ly6G (RB6-8C5), CD8α (53-6.7), B220 (RA3-6B2) and TER119 (TER-119) (all from BioLegend) and CD19 (1D3, BD Biosciences) (all at 1:200 dilution), and antibody-bound cells were removed with streptavidin-coated Dynabeads (ThermoFisher) using magnetic-activated cell sorting according to the manufacturer's instructions. Isolated T cells were rested overnight at 37 °C and 5% CO$_2$ at $1 \times 10^6$ cells/ml in R10 medium, which consisted of RPMI 1640 (Sigma) supplemented with 10% fetal calf serum (FCS), 100 U/ml penicillin, 100 μg/ml streptomycin, 2 mM L-glutamine, 1 mM sodium pyruvate, non-essential amino acids, 100 μM 2-mercaptoethanol (all Sigma-Aldrich) and 20 mM HEPES (Gibco).

## T cell migration under agarose

8-well polymer or glass chamber μ-slides (ibidi, 80821 or 80827) were coated with recombinant mouse ICAM-1-Human-Fc chimera (RnD, 796-IC) at 3 μg/mL in PBS overnight at 4 °C. Slides were washed 3x with PBS, blocked with 2% bovine serum albumin (BSA) (Sigma-Aldrich), and washed 3x again. Agarose gel preparation was adapted from a previously described method[47]. Agarose (0.5%, UltraPure Agarose, ThermoFisher) in R10-equivalent medium was prepared at 50 °C, and, if required, mouse CCL21 (250 ng/mL RnD, 457-6 C) and inhibitors [WNK463, (WNKi, MedChemExpress), bumetanide (SLC12Ai, 20 μM, Abcam), DFP00173 (AQP3i, 20 μM, Axon Med Chem), AER-270 (AQP4i, 0.25 μM, Tocris)] or equivalent amounts of vehicle control (DMSO, ThermoFisher) were added, and the mixture immediately plated onto the μ-slides. The agarose was cooled at 4 °C for 1 h, and then equilibrated at 37 °C and 5% CO$_2$ for at least 30 min. To change the osmolarity of the gel, part of the solution was replaced with NaCl-free HBSS to generate a gel of the required osmolarity. For isotonic agarose gels with reduced NaCl, the NaCl-free HBSS was supplemented with L-glucose (Sigma). Osmolarity was verified using an osmometer (Model 3250, Advanced Instruments).

T cells stained with CellTrace Violet (CTV, 1 μM, ThermoFisher) were resuspended in a small volume of R10 medium containing propidium iodide (1.5 μM) and injected under the agarose gel with a pipette. Cells were acclimatized at the microscope for at least 30 min in a humidified chamber at 37 °C and 5% CO$_2$ and then imaged every 15 s for 10 min using a Nikon Ti2 Eclipse microscope with an LED illumination system. Images were acquired with a Nikon 20x (0.75 NA) Ph2 plan apo objective, at 2048 × 2048 resolution in 16-bit using Micro-Manager[48]. Cells were tracked using the FIJI plugin TrackMate[49]. Propidium iodide-containing dead cells, and tracks shorter than half the video length were excluded. To determine cell shapes, binary images were produced from the CTV signal from one still frame per video stack using the FIJI 'triangle' auto-thresholding algorithm. Cell profiles were smoothed using the 'open' binary function and particles between 38 and 60 μm$^2$ were detected with the 'analyze particle' tool.

## T cell migration in a Collagen-I matrix

To study T cell migration in a collagen-I matrix a previously described method was adapted[50]. CTV-labeled CD4$^+$ T cells ($2 \times 10^6$ cells/ml) in R10 medium containing rat tail Collagen-I (1.5 mg/ml, ibidi) and CCL21 (250 ng/ml) were mixed on ice and plated into wells of a 15-well angiogenesis μ-slide (ibidi) and incubated at 37 °C and 5% CO$_2$ for 30 min to allow the collagen to polymerize. R10 medium containing CCL21, propidium iodide (1.5 μM) and inhibitors or vehicle control if required, was added to the wells and slides incubated in a humidified chamber on the microscope at 37 °C and 5% CO$_2$ for a further 30 min. Cells were imaged using a Zeiss LSM880 with an AiryScan detector, a 405 nm laser to excite both CTV and propidium iodide and a Zeiss 10x plan apo (0.45 N.A.) air objective. A z-stack consisting of 14 z-slices with an 8 μm spacing were acquired every 15 s for 5–10 min. Cells were tracked using the FIJI plugin TrackMate. Propidium iodide-positive cells and cells that were tracked for less than 1/3 of the video length were excluded from the analysis.

## Migration of activated T cells

Flat-bottomed multi-well cell culture plates (Corning) were incubated overnight at 4 °C with PBS containing plate-bound anti-CD3ε (4 μg/ml, 145-2C11, Tonbo Biosciences) and anti-CD28 (4 μg/ml, 37.51, BioLegend). Plates were washed with PBS and mouse naïve CD4$^+$ T cells were cultured in the wells at $1 \times 10^6$ cells/ml in R10 at 37 °C. On day 3 of activation, cells were transferred into fresh plates, stimulated daily with 20 ng/mL recombinant IL-2 (PeproTech) and used on day 5 of activation. Under agarose migration assays were run as described above, except using 250 ng/mL CXCL12 (R&D Systems, 460-SD-01/CF) instead of CCL21.

Chemotaxis assays were run according to the manufacturer's instructions (ibidi μ-slide chemotaxis, 80326) using CXCL12 as a chemoattractant. Briefly, cells were suspended in a 3D 1.5 mg/mL rat-tail collagen-I matrix in the observation area of the slide. Chemotaxis chambers reservoirs were loaded and cells were imaged for 1 h using a Nikon Ti2 Eclipse and a 20x (0.75 NA) Ph2 objective at 37 °C and 5% CO$_2$.

## Immunoblotting

Mouse naïve CD4$^+$ cells were rested in T cell media for >3 h. $2–4 \times 10^6$ cells per condition were preincubated with vehicle control or WNKi for 30 min at 37 °C. Cells were stimulated with 250 ng/mL CCL21 for the indicated amount of time. Cells were lysed in RIPA buffer (50 mM Tris, 150 mM NaCl, 50 mM NaF, 5 mM dithiothreitol, 2 mM EDTA, 2 mM Na$_4$P$_2$O$_7$, 1 mM Na$_3$VO$_4$, 1% Triton X100, 0.5% deoxycholate, 0.1%, sodium dodecyl sulfate (SDS), 5 mM dithiothreitol, 1× PhoStop (Roche), 1× complete protease inhibitor cocktail (Roche)) for 15 min on ice. Samples were pelleted at 16,000×G for 10 min at 4 °C. Lysates were diluted in SDS sample buffer (0.2 M Tris-HCl, 3% SDS, 10% glycerol, 3% 2-mercaptoethanol, 0.25% bromophenol blue) and heated at 95 °C for 5 min.

Lysates were analyzed by electrophoresis on 4–20% SDS-PAGE gels at 120 V for 1 h in Tris-Glycine-SDS running buffer. Separated proteins were transferred to methanol-activated PVDF membranes using Turboblot transfer (BioRad). Membranes were blocked for 90 min in Li-Cor Odyssey Blocking buffer. Primary antibodies (anti-pOXSR1 polyclonal, MRC-PPU Dundee; anti-ERK2, Cell Signalling Technologies, 9108) were diluted in Li-Cor buffer at 1:500 dilution and incubated overnight at 4 °C. Membranes were washed 3x in PBS + 0.1% Tween20 (PBST). Membranes were incubated with secondary antibodies [α-sheep IgG (H + L) AlexaFluor 680 and α-rabbit IgG (H + L) WesternDot 800 (both Invigrogen)] at 1:1000 at room temperature for 1 h, washed in PBST and imaged using Li-Cor CLx. Blots were analyzed on ImageStudio.

## Determination of element concentrations using inductively-coupled plasma mass spectrometry (ICP-MS)

ICP-MS samples were prepared using a protocol adapted from Stangherlin et al.[51]. T cells were rested overnight in R10 medium. Cells were pre-treated with inhibitor or vehicle in R10 for at least 20 min at 37 °C and 5% $CO_2$. Following pre-treatment, cells were stimulated with CCL21 (250 ng/mL final concentration) or medium only for 10 min at 37 °C and 5% $CO_2$. Cells were washed twice in ice-cold iso-osmotic wash buffer (300 mM sucrose, 10 mM Tris base, 1 mM EDTA in purified water, pH 7.4 adjusted with acetic acid [Fluka]). Cell pellets were digested in trace metal grade concentrated nitric acid (67–69% w/w) for 15 min at room temperature, diluted using purified water with a resistivity ≥18.2 MΩ cm from a Milli-Q system (Merck Millipore) and then spiked with either gallium (Ga) or cerium (Ce). Sample measurements were conducted at two institutes. At the London Metallomics Facility at King's College London, samples were analyzed on a Thermo Scientific iCAP TQ Inductively Coupled Plasma Quadrupole Mass Spectrometer (ICP-QMS) operating in Dynamic Reaction Cell (DRC) mode with oxygen as the reaction gas. The introduction system to the instrument was a Cetac ASX-520 autosampler coupled to a MicroMist nebulizer that was fitted to a quartz cyclonic spray chamber. Data reduction involved the normalization of raw intensities using Ga as an internal standard, then blank correcting the signal by removing the average analyte intensity of repeat blank measurements and applying external standardization using a six-point calibration curve to convert the corrected intensities into concentration measurements. At the Department of Earth Sciences, University of Cambridge, samples were analyzed using a Perkin Elmer Nexion 350D ICP-MS and analyte intensities were normalized based on cerium (Ce) (Romil, E3CE#). For each sample, element concentrations were normalized to cell numbers. Data from the two measurement sites were combined by normalizing element concentrations to unstimulated control cells.

## NaCl reduction and Cl replacement

To generate media of varying NaCl compositions, NaCl-free RPMI was supplemented with additional NaCl (Honeywell, S9888). To make the medium isotonic, the remaining difference in osmolarity (up to 300 mOsm/L) was made up using D-sorbitol (Sigma, S1876), an inert osmolyte. Osmolarity was verified using an osmometer (Model 3250, Advanced Instruments). To generate medium with reduced Cl concentration, NaCl-free RPMI was made isotonic through the addition of Na gluconate (Sigma, S2054-100G). Media was supplemented with 0.5% BSA and propidium iodide (1.5 μM). CTV-stained T cells were suspended in medium containing 250 ng/mL CCL21, allowed to adhere for 1 h at 37 °C and imaged on a Nikon Ti2 Eclipse widefield microscope using a 20x (0.75 NA) Ph2 objective at 37 °C and 5% $CO_2$.

## Cell volume measurement

T cells were rested overnight in R10 medium. Inhibitor or vehicle control was added for 20 min and then CCL21 or medium for a further 20 min. Cells were resuspended in CASYton fluid (Cambridge Bioscience) and median cell volumes measured using the Coulter counter principle on a CASY cell counter (Cambridge Bioscience).

Alternatively, cell volume was determined using microscopy. Naïve CD4+ T cells purified from ROSA-mTmG mice expressing tdTomato-CAAX to visualize the plasma membrane were imaged in glass 8-well μ-slides (ibidi, 80827) coated with ICAM-1 (3 μg/mL) in phenol red-free RPMI supplemented with 10% FCS. Cells were treated with WNKi or vehicle control and stimulated with 250 ng/mL CCL21 and rested on the microscope for 30 min prior to imaging. Cells were imaged using instant structural illumination microscopy (iSIM) using an iSIM microscope (VisiTech) with an Olympus 150x TIRF apo (1.45 NA) oil-immersion objective. Cells were excited using 552 nm wavelength laser, with emission detected using an sCMOS camera (Prime BSI Express, Teledyne Photometrics). Cells were either imaged for

2 min every 15 s or acquired as a still frame. Resulting images were deconvoluted using Microvolution algorithms in μ-Manager. Cell volume was calculated by generating a mask of plasma membrane edges and summing up the areas of individual z-slices in each cell measured using the 'analyze particles' function in FIJI.

## $^2H_2O$ uptake assay

T cells were rested overnight in R10 medium. Cells from one biological replicate were divided into four conditions and pre-treated with WNKi or vehicle in $H_2O$-based R10 for at least 20 min at 37 °C and 5% $CO_2$. Following pre-treatment, cells were stimulated with 250 ng/mL CCL21 in $^2H_2O$-based R10 (deuterium oxide, Sigma-Aldrich, 435767) containing vehicle or WNKi. Cells were washed twice in ice-cold $H_2O$-based PBS and lysed using lysis buffer (50 mM HEPES, 150 mM NaCl, 1% NP40 in ultrapure $H_2O$ containing 1:500 uniformly $^2H$-labeled 1,4-dioxane-d$_8$ [Sigma Aldrich, 186406] as an internal standard) for 20 min at room temperature. Lysate supernatants were measured using nuclear magnetic resonance (NMR) spectroscopy (Bruker Advance III HD 700 equipped with a 5 mm QCI cryoprobe) at 25 °C employing 3 mm NMR tubes. Fully-relaxed one-dimensional $^2H$ spectra were recorded in the absence of field-frequency lock using a 30° excitation pulse, 8 s relaxation delay, 20 ppm sweep width, 1.6 s acquisition time, 2 dummy scans and 16 accumulated transitions (total experiment time 3 min per sample). Data processing in Bruker Topspin 3.6 employed zero-filling to 128 K points, 4 Hz line-broadening, Fourier transformation and baseline correction. For each sample, the peak integral of the $^2H$ resonance of $^2H_2O$ was normalized to the that of the $^2H$-labeled 1,4-dioxane.

## Catalase treatment

To determine the efficacy of catalase treatment, mouse naïve CD4+ T cells were loaded with DCFDA (ab113851, Abcam) according to the manufacturer's protocol. Cells were pre-treated with bovine liver catalase (C40, Sigma Aldrich) for 30 min at the indicated concentrations. Cells were treated with 1 mM $H_2O_2$ for 30 min at room temperature, and DCFDA fluorescence was measured by flow cytometry (LSRFortessa X-20, BD Biosciences).

For migration experiments, overnight rested mouse naïve CD4+ T cells were pre-treated with 20 U/mL catalase in R10 for at least 30 min. Cells were injected under agarose containing 20 U/mL catalase and CCL21 in 8-well dishes coated in ICAM-1, as described in other experiments. Cells were rested for 30 min at the microscope prior to imaging as described above.

## Generation of CD4+ T cells expressing GFP-WNK1, GFP-OXSR1 and GFP-SLC12A2

DNA fragments encoding open reading frames for GFP-WNK1, GFP-OXSR1 and GFP-SLC12A2 were inserted into the MIGR1 retroviral vector[52], replacing the IRES-GFP sequence, such that fusion proteins were expressed from the 5' LTR of the vector. In all cases the eGFP was fused in-frame to the N-terminus of the human WNK pathway proteins. The GFP-WNK1 and GFP-SLC12A2 constructs were generated by the MRC Protein Phosphorylation Unit. All constructs were validated by DNA sequencing.

Plasmid DNA for MIGR1 vectors expressing GFP-WNK1, GFP-OXSR1 or GFP-SLC12A2 and MIGR1 expressing GFP only were transfected into PlatE cells[53] to generate medium containing infectious retrovirus. This was used to infect bone marrow cells from C57BL/6J mice that were in turn used to reconstitute irradiated (5 Gy, $^{37}$Cs source) RAG1-deficient mice[54].

## Live cell imaging of T cells expressing WNK1 pathway proteins

CD4+ T cells expressing GFP-tagged WNK1 pathway proteins were purified from lymph nodes of bone marrow chimeras by flow cytometric sorting for GFP+ T cells using an Avalon sorter (Propel Labs/Bio-

Rad). Sorted cells were incubated overnight in R10 medium at 37 °C and 5% $CO_2$ and then injected under agarose in an 8-well plastic bottom μ-slide containing CCL21 as described earlier. Cells were placed in a humidified chamber at 37 °C and 5% $CO_2$ on a Zeiss LSM 880 inverted microscope for 30 min, and then imaged by confocal microscopy with a 63x plan apo (1.4 N.A.) oil-immersion objective. Cells were imaged every 15 s for 5 min in a single z-plane proximal to the coverslip. To determine the localization of GFP or GFP-tagged proteins in migrating T cells we used the FIJI plugin VirusTracker[55]. The plugin detects the direction of migration and was set to capture a 20 μm-long region of interest along the length of the cell. Images of these cell shapes were projected on top of each other using the mean intensity projection option.

To image nuclear-excluded GFP, mouse naïve CD4[+] T cells were transfected with p8xEGFP-N1 (Addgene # 122168, gift from Georg Mayr) which expresses 8 concatemeric EGFP proteins, resulting in nuclear exclusion. Cells were electroporated using the X-001 program on a Nucleofector (Lonza) using a mouse T cell nucleofection kit (Lonza, VPA-1006), and rested overnight in R10 medium. EGFP[+] T cells were sorted and incubated in R10 medium for at least 2 h at 37 °C and 5% $CO_2$. Cells were injected under agarose containing CCL21 and allowed to acclimatize for at least 30 min in a humidified chamber on the microscope at 37 °C and 5% $CO_2$. Cells were imaged using a Leica SP8 FALCON confocal inverted microscope, equipped with a 63x plan apo (1.4 NA) oil-immersion objective every 15 s for 2.5 or 5 min. Cells were excited using a white light source at 488 nm for EGFP. To determine protein localization, cells were aligned in the direction of migration and a 14.4 × 7.5 μm box was drawn around the outline of the cell. Images of these cell shapes were projected on top of each other using the mean intensity projection option.

### Imaging of WNK1 pathway proteins in fixed T cells
15-well angiogenesis or 18- well μ-slides (ibidi, 81506 or 81816) were coated with 3 μg/ml mouse ICAM-1-human Fc fusion protein overnight at 4 °C. Slides were washed 3x with PBS, blocked with 2% BSA in PBS for 20 min at room temperature, and washed again three times. Naïve CD4[+] T cells that had been incubated overnight at 37 °C were resuspended in RPMI containing 0.5% BSA, added to the μ-slides, and allowed to adhere for 1 h at 37 °C. Cells were stimulated with 250 ng/ml CCL21 for 20 min, and then fixed in 4% paraformaldehyde for 15 min at room temperature. Cells were washed with PBS, blocked in 1% BSA in PBS (blocking buffer) for 20 min, permeabilized with 0.1% Tween20 in PBS for 15 min, washed with PBS and stained in blocking buffer overnight at 4 °C with anti-CDC42 (B-8 mouse monoclonal, Santa Cruz), biotin-conjugated anti-CD44 (IM7 rat monoclonal, BioLegend) and one of these primary antibodies: anti-p-WNK1 (S382), anti-OXSR1, anti-p-OXSR1 (S325), anti-SLC12A2, anti-p-SLC12A2 (T203/T207/T212) (all sheep polyclonal sera from MRC Protein Phosphorylation Unit) or anti-AQP3 (NBP1-97927 rabbit polyclonal, Novus Biologicals), all diluted at 1:200. Cells were washed with PBS and incubated in blocking buffer for 1 h with Streptavidin-AlexaFluor 488 (S11223, Molecular Probes), goat anti-Mouse IgG-AlexaFluor 555 (A-21424), and either goat anti-rabbit IgG-AlexaFluor 647 (A-21245) or donkey anti-sheep IgG-AlexaFluor 647 (A-21448, all from ThermoFisher), all diluted at 1:250. Cells were washed twice with 0.1% Tween20 in PBS and once with PBS containing DAPI to visualize the nucleus. Cells were washed once and covered with VectaMount AQ aqueous mounting medium (Vector Laboratories) and stored at 4 °C until imaging.

Cells were imaged on a Leica SP8 FALCON inverted confocal microscope with a 63× plan apo (1.4 NA) oil-immersion objective. Fluorophores were excited with a white light laser. Cells were imaged in z-stacks consisting of 7–9 z-slices with a 1 μm spacing. To determine localization of WNK1 pathway proteins, CDC42 and CD44 were used to determine the leading and trailing edges (LE and TE) respectively. A line the full width of the cell was drawn along the length of the cell from

the LE to the TE. Average fluorescence intensities were determined for each of 12 bins along the cell length. Intensity profiles were normalized from 0–1, where 0 was the lowest signal and 1 the highest signal per protein, per cell. Binned normalized fluorescence intensities were used to calculate the correlations between LE and TE markers and WNK1 pathway proteins. To evaluate polarization of WNK1 pathway proteins, fluorescence was measured along the perimeter of the cell and the polarization ratio was calculated by dividing the fluorescence in the 25% of the perimeter centered on the maximum fluorescence by the total fluorescence along the perimeter of the cell.

### Flow cytometric analysis of F-actin
Overnight-rested mouse naïve CD4[+] T cells were placed in RPMI, 0.5% BSA and stimulated with CCL21 for the required time. Cells were immediately fixed in 4% paraformaldehyde for 15 min. Cells were washed with PBS, permeabilized with 0.1% Tween20 in PBS for 15 min, and blocked for 1 h in 2% BSA in PBS. Cells were stained with phalloidin conjugated to iFluor-488 (Abcam) in blocking buffer for 1 h at room temperature, washed twice with 0.1% Tween20, and once with PBS. Cells were analyzed on an LSRII flow cytometer (BD Biosciences) and mean fluorescence intensity was quantified using FlowJo (BD Biosciences).

### Analysis of plasma membrane to F-actin spacing
Overnight-rested mouse naive CD4[+] T cells constitutively expressing LifeAct-eGFP were labeled with CellMask Orange (ThermoFisher) to stain the plasma membrane following the manufacturer's instructions. Cells were washed once with PBS, resuspended in R10 medium and injected under CCL21- and inhibitor-containing agarose in a glass bottom 8-well μ-slide dishes (ibidi) pre-coated with ICAM-1. Cells were imaged by iSIM using an iSIM microscope (VisiTech) with an Olympus 150x TIRF apo (1.45 NA) oil-immersion objective. Cells were simultaneously excited at 488 nm for LifeAct-eGFP and 561 nm for CellMask Orange. Emission fluorescence was split with a dichroic mirror with a long-pass filter at 561 nm and was simultaneously detected using 2 sCMOS cameras (Prime BSI Express, Teledyne Photometrics). Cells were imaged every 250 ms for a total of 30 s in a single z-plane proximal to the coverslip. To correct for chromatic aberration and camera alignment, 0.5 μm TetraSpeck fluorescent beads (ThermoFisher) were imaged after each session using the same acquisition settings as the cells.

Images were deconvoluted using Huygens software (Scientific Volume Imaging) with the 'Normal' deconvolution template. A random frame from the time-lapse series was chosen while viewing only the plasma membrane stain to avoid bias. The leading edge of the cell was determined from the direction of migration. Adapting a previously described analysis protocol[56], a 30-pixel wide line was drawn along the leading edge, with the plasma membrane placed in the middle of the line. The line and the data within it were straightened out and 20-pixel wide overlapping lines were drawn perpendicular to the plasma membrane signal, along the length of the leading edge. Fluorescence intensity data were normalized from 0-1, and the intensity profiles in each 20-pixel wide line were manually aligned according to the peak of plasma membrane fluorescence intensity. The distance between the peak plasma membrane signal and the peak F-actin signal was calculated within each line, with the peak of F-actin defined as the highest LifeAct-eGFP fluorescence intensity within 400 nm of the plasma membrane.

### Analysis of membrane proximal F-actin
Mouse naïve CD4[+] T cells were transfected with C1-MPAct-mCherry (Addgene #155222, gift from Tobias Meyer) and C1-eGFP-CaaX (Addgene #86056, gift from Lei Lu) plasmids using the X-001 program on a Nucleofector (Lonza) and rested overnight in T cell medium. GFP[+]mCherry[+] T cells were sorted and incubated in R10 medium for at

least 2 h at 37 °C and 5% $CO_2$. Cells were injected under agarose containing CCL21 and inhibitors or vehicle control and allowed to acclimatize for at least 30 min in a humidified chamber on the microscope at 37 °C and 5% $CO_2$. Cells were imaged using a Leica SP8 FALCON confocal inverted microscope, equipped with a 63x plan apo (1.4 NA) oil-immersion objective every 15 s for 2.5 or 5 min. Cells were excited using a white light source at 488 nm for eGFP-CaaX and at 585 nm for MPAct-mCherry, and acquired simultaneously.

To analyze the distribution of the MPAct probe, one random frame per migrating cell was selected using FIJI. Bias was avoided by selecting frames using only the GFP-CaaX signal. The leading edge of the cell was determined from the direction of migration. A line the full length and width of the cell was draw from the front to the back of the cell. Average fluorescence intensities were determined for each of 12 bins along the cell length. Fluorescence intensity levels were normalized with the highest intensity value per fluorophore, per cell set to 1. The normalized MPAct-mCherry signal was divided by the normalized eGFP-CaaX signal to generate the MPAct/CaaX ratio. Polarization of the MPAct signal was determined by dividing the MPAct/CaaX ratio in the front 3 bins by the ratio in the rear 3 bins.

### TIRF microscopy of F-actin retrograde flow
35 mm round glass-bottom dishes (Greiner Bio-One) were cleaned with 1 M HCl for 15 min and washed three times with water. Dishes were plasma-activated for 30 s (HPT-100, Henniker Plasma) and coated with 0.2 mg/ml poly(L-lysine)-graft-PEG (SuSoS) in PBS overnight at 4 °C, blocked with 2% BSA in PBS for 20 min, and washed three times with PBS. Overnight-rested mouse naïve $CD4^+$ T cells constitutively expressing LifeAct-eGFP were injected under agarose containing CCL21 and either inhibitors or vehicle control. Cells were imaged at 37 °C and 5% $CO_2$ using an Olympus IX83 total internal reflection fluorescence (TIRF) microscope with a 150 × 1.45 NA TIRF apo oil-immersion objective every 1 s for 1 min. Actin flow rates were determined from kymographs generated using FIJI.

### RNA sequencing data
Data on expression of Aqp and Slc12a mRNAs in mouse $CD4^+$ T cells was taken from an RNA sequencing analysis of naive mouse $CD4^+$ T cells (Sequence Read Archive, SRA accession number SRP059425)[6].

### Statistical analysis
All measurements were made on distinct samples (e.g. cells). No sample was measured repeatedly. To determine statistical significance between two unpaired groups, we used a 2-sided Mann–Whitney U test. To determine statistical significance when there were more than two groups and one variable, we used either a Kruskal-Wallis test when the data was not normally distributed, and unpaired and unmatched, or a 1-way ANOVA when the data was normally distributed and unmatched. When comparing multiple paired groups, we used a Friedman test. When comparing two or more groups, but with two variables (i.e. time and inhibitor/genotype) a 2-way ANOVA or a mixed effects analysis was used. To correct $p$-values for multiple comparisons we used the False Discovery Rate (FDR) method with the two-stage step-up to generate $q$-values[57, 58]. $q < 0.05$ was used as a cut-off to determine statistical significance. In cases where there was no multiple comparison correction (Mann–Whitney tests), $p < 0.05$ was used as a cut-off to determine statistical significance. All statistical analysis was performed using Prism (GraphPad). $p$- or $q$-values are quoted as ranges as indicated in the Figure legends: *$0.01 < p$ or $q < 0.05$, **$0.001 < p$ or $q < 0.01$, ***$0.0001 < p$ or $q < 0.001$, **** $p$ or $q < 0.0001$.

### Reporting summary
Further information on research design is available in the Nature Portfolio Reporting Summary linked to this article.

### Data availability
Mouse strains and plasmids available on request. Other data that support the findings of this study are available from the corresponding author upon reasonable request. RNA sequencing data used in this study have been previously published and can be found under SRA accession number SRP059425[6]. Source data are provided with this paper.

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

## Acknowledgements

We thank Erik Sahai and Michael Way for critical reading of this manuscript. We thank Rachel Edgar and John O'Neill for helpful discussions. We thank Miriam Llorian Sopena for help with analysis of RNA-seq data. We thank the Advanced Light Microscopy, Flow Cytometry and Biological Research Facilities of the Francis Crick Institute for microscopy, flow cytometry and for animal husbandry. We thank Chou-Long Huang, Dario Alessi and Sung-Sen Yang for mouse strains. We thank Rachel Toth for generation of plasmids, and the MRC Protein Phosphorylation Unit for antibodies. V.L.J.T. was supported by the Francis Crick Institute which receives its core funding from Cancer Research UK (CC2080), the UK Medical Research Council (CC2080), and the Wellcome Trust (CC2080), and by a grant from UKRI Biotechnology and Biological Sciences Research Council (BB/V0088757/1). This work was partly supported by the Francis Crick Institute through provision of access to the MRC Biomedical NMR Centre. L.L.d.B. was funded by an Imperial College London President's PhD Scholarship. For the purpose of Open Access, the author has applied a CC-BY public copyright licence to any Author Accepted Manuscript version arising from this submission.

## Author contributions

Conceptualization: L.L.d.B, V.L.J.T. Data curation: LLd.B. Formal analysis: LLd.B., S.M. Funding acquisition: V.L.J.T., L.L.d.B. Investigation: L.L.d.B, S.M., L.V., D.H., J.B.O.M., P.D., J.D., A.G., R.M., A.M., J.M. Methodology:

L.L.d.B., R.K. Project administration: V.L.J.T. Resources: J.B.O.M., R.K., D.H., L.V. Supervision: V.L.J.T., L.L.dB. Visualization: L.L.d.B. Writing – original draft: L.L.d.B., V.L.J.T. Writing – review & editing: L.L.d.B., V.L.J.T.

## Funding

## Competing interests
The authors declare no competing interests.
