## [Peer Review File · Nature Communications]

T cell migration requires ion and water influx to regulate actin polymerizationREVIEWER COMMENTS

Reviewer #1 (Remarks to the Author):

These investigators previously demonstrated that the migration of T cells depends on the WNK1 kinase, its downstream kinases STK39 and OXSR1 (also called SPAK and OSR1), and the ion transporter regulated by SPAK and OSR1, the sodium-potassium-2-chloride cotransporter SLC12A2 (also called NKCC1). Here, they follow up on those studies to understand the mechanisms by which this kinase/ion transporter pathway may be controlling cell migration. They demonstrate decreased T cell migration in response to CCL21 in ex vivo assays when the activity of WNK1, SPAK/OSR1, or NKCC1, as well as the water channel AQP3, are decreased via genetic or pharmacological manipulations. The activated (phosphorylated) forms of WNK1, OSR1, and NKCC1, as well as total OSR1 and AQP3, are preferentially found at the leading edge of migrating T cells, where they colocalize with CDC42. They hypothesize that ion and water influx through NKCC1 and AQP3, regulated by the WNK-SPAK/OSR1 kinase cascade, increases local ion and water influx to generate space between F-actin and the plasma membrane, in order to add actin to the leading edge F-actin and allow migration via the Brownian ratchet mechanism. This is examined using cell biological and physiological approaches.

Overall, this is an attractive hypothesis that may be relevant not only to T cell migration, but also to other cell types. Additional experiments could further bolster the findings. For example, all the data on NKCC1 and AQP3 rely on a single pharmacological inhibitor for each transporter/channel. The Scl12a2 knockout mice are viable, and in fact, the authors used T cells from Slc12a2 knockouts in their previous paper, and saw effects similar to WNK1 knockdown or SPAK/OSR1 double knockdown, unlike in this paper in which they consistently see partial effects of bumetanide. Although they argue that this could be due to redundant effects of four other SLC12A transporters, no information is provided about which of these are expressed in T cells, nor is any attempt made to experimentally test this. This is important since their model is that water moving through AQP3 follows ion influx through ion transporters. Another approach to testing the role of NKCC1 would be to omit extracellular chloride. Genetic manipulation of AQP3 would also strengthen the findings using the AQP3 inhibitor, eg using T cells harvested from Aqp3 knockout mice (which are viable). Another question is whether the WNK1-SPAK/OSR1 kinase cascade is regulating AQP3, since AQP3 inhibition fully recapitulates the phenotype observed with loss of WNK1 or SPAK/OSR1. This could be examined in heterologous cells.

The authors demonstrate that mild hypotonicity rescues the migration defect in T cells treated with a very low concentration (0.2 μ M) of WNK inhibitor (Fig 5C). They get a smaller degree of rescue in isotonic medium with low NaCl (Fig 5D). A known consequence of hypotonic exposure is a decrease in intracellular chloride due to the regulatory volume decrease response. Since chloride is a negative regulator of WNKs, and a very low concentration of WNK inhibitor is being used in this experiment, one possibility is that the hypotonic medium is allowing increased activation of the residual WNK, rather than the water influx per se explaining the rescue. Thus, converse to the third set of bars in this figure, it would be important to lower osmolality while keeping chloride the same (or even high) to rule out this possibility. This could be done by using L-glucose in the isotonic medium, and omitting it for the hypotonic. Simply lowering extracellular chloride without changing tonicity may not be sufficient to lower intracellular chloride, depending on the chloride conductance of these cells, so the experiment performed (ie isotonic low NaCl) does not rule out a possible intracellular chloride effect. Examining the degree of WNK1 S382

phosphorylation (reflective of WNK autophosphorylation and activation) under the various conditions could also lend insight into this question.

Minor:

1. It's a bit difficult to follow the combination of genetics and bone marrow chimeras by reading the Methods/Results/Figure legends. It would be helpful to have a supplemental table in which this is detailed for the figures using these, ie, genotype, when tamoxifen is applied, which bone marrow cells are transferred into which mice, and where the T cells are harvested from, as well as abbreviations used (eg, "RCE" = ROSA26 CreERT2). Similarly, "CTV" appears in the video/figure legends without being spelled out; readers should not have to go to the Methods to figure out what this non-standard abbreviation stands for. It would be helpful to indicate in Figure 1C that "WNKi" = "WNK inhibitor (WNK463)", and in 1D that "SLC12A2i" = "NKCC1 inhibitor (bumetanide)" and "AQP3i" = "AQP3 inhibitor (DFP00173)."
2. The authors note in the discussion that SLC12A2 inhibition reduces migration of glioblastoma cells. In fact, WNK1 or OSR1 knockdown also inhibits glioblastoma cell migration (Zhu, Molecular Cancer 2014, 13:31), further supporting the authors' speculation that the mechanism they describe in T cells may also be relevant to other cell types.

Reviewer #2 (Remarks to the Author):

The authors investigate mechanisms by which WNK1 kinase and the cascade it regulates related to SLC12A2 and water entry through AQP3 creates space for F-actin polymerization at the leading edge of migrating T cells. 3D Motility of tumour cell by blebbing is accepted to require contraction to drive cytoplasm into blebs ahead of F-actin polymerization. There is also a hydraulic engine model for motility that proposes forward movement by osmosis that is secondarily supported by F-actin. That authors propose that water entry through coupled ion transport by SLC12A2 and water transport by AQP3 supports advancing the leading edge ahead of F-actin polymerisation. The authors build a case for this model through a series of experiments focused on polarization of the key molecular players in the cascade, the location of F-actin in relation to the plasma membrane, and manipulation of ions and osmotic gradients. The authors give only passing mention of alternative models that could be quantitatively tested against the authors preferred hypothesis.

1. In Figure 1, how long were the different inhibitors applied. Its striking that WNKi reduced cell volume in a dose dependent manner similar to the effect of the KO, whereas SLC12A2i and AQP3i didn't alter cell volume. Were different times of exposure involved. The WNKi is more potent at blocking migration. Could concentrations of the WNKi and AQP3i that generate similar effects on motility be compared? This may result in a fair comparison that may not support polarization of these signals in relation to the distribution of cytosol. The AQP3i inhibit all water channels or just AQP3? If inhibiting all water channels shouldn't the effect be greater?
2. A complication with Figure 2 is that WNK1 and OSR1 appear excluded from the nucleus, whereas GFP is not, so most of the effect in C may be related to nuclear exclusion and polarization of cytoplasm. Can the authors use a form of GFP that is excluded from the nucleus as a control? Similarly, for SLC12A2, a membrane protein, GFP is not an appropriate control. The CAAX construct used later might be more suitable, although this is potentially complicated if SLC12A2 is not confined to the plasma membrane.

3. Figure 3 interprets a separation of the average F-actin location from the average membrane location as evidence for a water filled gap between F-actin and the membrane. There are two weaknesses of this approach. First, we know that the lymphocyte surface has sub-resolved membrane protrusion that contain F-actin, in addition to cortical f-actin and the extensive branched F-actin network in the leading edge that is expected to have a include F-actin mass that is well separated from the membrane. So the authors would need to consider other models that can generate the same separation and somehow rule these out. The genetics of the system also seems to impact the extent of dendritic nucleation of F-actin and this could be the major effect. The other issue is that building a reasonable model would require information on membrane topology that is not resolved. They would need a good way to go from the averaged data to a model of membrane topology for the lateral and leading-edge structures under different conditions to generate a model. This would require electron tomography or at least SIM level super-resolution data, if not dSTORM or expansion microscopy. The observation that perturbations that alter motility also alter this measurement could have many explanations that don't involve a water filled space into which F-actin polymerizes.
4. In Fig 4 the system used appears to confirm that inhibiting these pathways (WNK1, SLC12A2 and AQP3) reduce the degree of dendritic F-actin, which based on the relationship of the C-actin to the membrane would reduce the signal in this experiment. I don't think this related to the creation of space, but more to the fraction of F-actin that is in range of the membrane anchored F-acting ligand.
5. In Figure 5 the authors apply the approach I suggested above and use the WNK1i at a 0.2 μM to allow polarization with effects more similar to the SLC12A2 or AQP3 KO/inhibitor studies. However, even with this approach, it is not clear that the results are as they would predict. Reducing tonicity would just non-specifically swell the cells through all aquaporins and this restored motility and makes the motility independent of weak inhibition of WNK1i. What happens if a higher concentration of WNKi or the knockout? Does the low tonicity rescue motility and F-actin flow? Are there situations where effects are reverses and WNKi stimulations mobility?

I worry that there are other biophysical and biochemical effects that lead to collapse of polarity with WNK1i, knockout and its restoration by manipulation of cytoplasmic ion levels. Certainly, effect on the order seen with AQPi could be based on the proposed water entry promoting membrane protrusion, but the entire effect of WNK1 loss or inhibition seems well beyond what can be accounted for by any osmotic effects. It should also not be forgotten that WNK1 deficient cells become hyperadhesive, from earlier work from this lab, which also requires actin-based contractility to activate LFA-1. How does this relate to this model? Can the cells spread and generate F-actin based contractility in the interface? There are also NaCl dependent kinases in the cells that could also provide a mechanism for effects of NaCl entry.

Reviewer #3 (Remarks to the Author):

The manuscript by de Boer et al. builds on a previous study by the authors on the identification via an RNAi screen of WNK1 as a negative regulator of integrin-mediated adhesion and a positive regulator of chemokine-driven migration in T cells (Köchli R. et al., Nat Immunol 2017). As it is the case in other cellular models, the authors had found that downstream of the chemokine receptor CCR7, WNK1 activates the OXSR1 and STK39 kinases and the SLC12A2 ion co-transporter. These previous data were suggestive of the

fact that ion influx is required for T cell migration.

In the current study, the authors propose to investigate how the WNK1-SLS12A2 axis regulates T cell migration. They report that deficiency/inhibition or expression of mutant of WNK1, OXSR1, STK39 and SLS12A2 resulted in decreased migration of murine CD4⁺ T cells under confinement (under-agarose, 3D collagen-I matrix). This piece of data consolidates data from their previous study. They then show the polarization of WNK1, OXSR1 and SLS12A2 at the leading edge of T cells migrating in confined in vitro settings. This suggests that CCR7 signaling via WNK1, OXSR1 and STK39 leads to localized entry of Na⁺, K⁺ and Cl⁻ ions via SLC12A2 at the leading edge of migrating T cells. The authors then report that CCL21 induced a volume increase in WNK1-expressing T cells, whereas no such increase was seen in the absence of WNK1, or if WNK1 was inhibited. This supports the hypothesis that WNK1 pathway activity results in water influx, which is required for volume regulation. As they report, this hypothesis aligns with the concept of the « osmotic engine model » (Stroka, K.M. et al. Cell 2014). Then, the authors focus on AQP3 as a candidate aquaporin downstream the WNK1-SLS12A2 axis. In line with the hypothesis of the authors, AQP3 inhibition caused a significant decrease in the speed of CCL21-induced migration of T cells under agarose or in a collagen-I matrix. The authors then employ structured illumination microscopy and previously validated fluorescent probes to explore the impact of the WNK1-SLS12A2-AQP3 axis on the spacing between the leading edge plasma membrane and the underlying actin cytoskeleton. Treatment of T cells with inhibitors targeting WNK, SLC12A2 or AQP3 resulted in an increase in the membrane-proximal actin distance at the leading edge, implying that this axis via polarized osmotic uptake of water might favor the actin polymerization process driving cell protrusion and migration. To complement these observations, the authors measure retrograde actin flow to show that it is reduced in WNK1-deficient T cells or T cells treated with inhibitors of WNK1, SLC12A2 or AQP3. Finally in order to more directly test the contribution of water influx in the observed process, they modulate the tonicity of the medium to report increased migration speed in cells where WNK1 had been partially inhibited.

Overall this is an interesting in vitro study addressing very basic aspects of polarized T cell migration. Although the investigated pathway is not novel, the study provides some new exploration to connect chemokine sensing to polarized migration via a conserved pathway involving ion transport and osmotic regulation. The manuscript is well written and the data are presented in clearly organized figures. Some of the conclusions that the authors formulate are however to far stretched as they go beyond what is demonstrated through the provided experimental work. As explained below, further work would be required to properly address some of the key points the authors try to make in this study.

Major points :

1- Possibility that influx of H₂O₂ rather than (or together with) H₂O would explain the role of the WNK1-SLC12A2-AQP3 axis in regulating leading-edge actin remodeling to drive polarized T cell migration.

The current study agrees with a previous publication reporting that AQP3-deficient T cells have defective migration in response to various chemokines both in vitro and in vivo (Hara-Chikuma, M. et al. Chemokine-dependent T cell migration requires aquaporin-3-mediated hydrogen peroxide uptake. J Exp Med 209, 1743-52, 2012). This previous publication however showed a major role of AQP3 for hydrogen peroxide uptake, which was found to

activate Cdc42 and polarized actin remodeling via WASP and ARP2/3. Since the authors defend here an alternative mechanism (water influx facilitating actin remodeling by creating space), it is essential that they decipher the relative contribution of water versus H₂O₂ driven mechanisms underlying the phenotype of AQP3-deficient or AQP3-inhibited T cells. The authors would need to inhibit H₂O₂ transport to study the contribution of water influx to the phenotype (migration and leading edge actin remodeling). It is also expected that the authors introduce what the study by Hara-Chikuma, M. et al. showed and discuss the relative H₂O/H₂O₂ contribution.

2- Measuring ion transport and water influx

While the reported experiments (deletion or inhibition of SLC12A2 and AQP3, use of hypotonic medium) are suggestive of a contribution of ion transport and water influx in promoting actin remodeling of T cells stimulated with CCL21, direct measurements of such fluxes are missing. Such measurements would be quite important to sustain the claims made by authors in their manuscript.

In relation to this comment, the following sentences need to be revised according to what is precisely demonstrated experimentally:

“chemokine receptor-induced activation of the WNK1 pathway leads to water influx, most likely via AQP3, which is required for actin polymerization and migration”

“this chemokine receptor-induced water entry generates increased spacing between the plasma membrane and the underlying actin cytoskeleton at the leading edge”

“WNK1 activity and ion and water influx are required to maintain reduced MPA at the leading edge of T cells”

“Thus, WNK1 and ion and water influx are required for actin retrograde flow”

3- Resolution of the F-actin-membrane distance measurements

The authors claim they want to investigate the fluctuating gap separating the membrane and the actin filaments, which is a 10nm range phenomenon, with structured illumination microscopy. However, such approach is not expected to provide a sufficient resolution. The plasma membrane to “peak” F-actin signal that the authors measure seem to correspond to the distance between the plasma membrane and an inner part of the lamellipodial protrusion (for leading edge measurement) or to the core of the cortical actin (for the cell side measurements). While the reported measurements seem to indicate an effect the tested inhibitions, they might only reflect an overall change in actin cytoskeleton organization. Differently, the MPAct/CaaX ratio measurements appear to be a more adapted method to measure of the distribution of membrane-proximal actin. I would therefore advise to focus on this latter approach.

In addition to the resolution issue related to the experiments conducted with SIM, the images presented in Figure 3 do not display the expected distribution of F-actin at the polarized T cell leading edge, nor the expected organization of cortical actin on the cell periphery. The staining appears to be too weak and very patchy. The penetrations of LifeAct staining across the membrane-orange staining are also unexpected. It would be important for the authors to validate the quality of the stainings, e.g. by costaining fixed samples with Phalloidin.

4- Impact of the WNK1-SLC12A2-AQP3 axis on directional migration along chemokine gradients ?

While the provided experiments provide solid evidence for the role of the WNK1-SLC12A2-

AQP3 axis in CCL21-evoked T cell migration, they do not directly address directional migration along chemokine gradients. It is felt that cell tracks, cell shape and actin measurements in the context of chemokine gradients would reinforce the current study and would help understanding the precise contribution of this axis to chemokine sensing versus cell polarization versus orientation along gradients.

5- Which T cell motility steps are primary governed by the WNK1-SLC12A2-AQP3 axis ?

The authors had previously reported that WNK1 is activated downstream of CCR7, LFA-1 and the TCR (Köchli R. et al., Nat Immunol 2017). A question of central physiological relevance is therefore whether the WNK1-SLC12A2-AQP3 axis controls to a similar degree steps of T cell migration known to be distinctly balanced by chemokine receptors, integrins and the TCR, namely trans-endothelial migration, interstitial migration and antigen-presenting cell scanning. Complementary in vivo and in vitro experiments would help addressing this point. In particular, it is well accepted that CCR7 and the TCR play rather opposite roles in controlling T cell motility (polarized migration versus stop of migration). The authors could try unraveling how the addition of a TCR trigger to that of CCR7 alters the degree of WNK1 activation and possibly the downstream axis.

6- WNK1-SLC12A2-AQP3 axis in T cell subsets ?

Another direction that could greatly raise the interest for a larger community of immunologists would be to assess protein expression related to the explored pathway (WNK1, OXSR1, STK39, SLC12A2, AQP3) in various T cell subsets and under distinct activation/differentiation conditions. Indeed various T cell subsets (Th1 vs Th2, effector T cells vs T regulatory cells, resting versus activated T cells) display distinct motility properties. While there has been an obvious focus on trying to relate this to the expression of specific chemokine receptors and adhesion receptors, much less is known on the possible tuning of the conserved intracellular pathways that govern motility, such as the one studied here. Therefore protein expression study completed with a few key functional assays with drugs (as done on naïve CD4+ T cells in the manuscript) might reveal very interesting aspects of motility regulation underlying the properties of T cell subsets.

Further points requiring clarification (from figures):

Figure 1H : increase in cell vol upon CCL21 is very limited. It is difficult if not impossible to assess effects (or lack of effect) of inhibitors under such settings.

Figure 2B,C,E : indicate nb of cells (to make it clearer that this comes from cell averaging). Why are C and E represented differently (where is dispersion of individual cells)?

Figure 3 : normalization of intensity values: in B both membrane and actin seem to be normalized with max value = 1. In C only membrane seems to be normalized.

Figure 4: There is experimental variability of MP Act/Caax profiles along the cell length for vehicle when one compares C and D panels. This variability seems to be as important as the variations noted between the test conditions (WNKi, SLC12A2i, AQP3i). Please provide further evidence for the robustness of this assay and for the solidity of the data.

RESPONSE TO REVIEWERS' COMMENTS

A detailed response to the Reviewers' comments follows below, but for clarity, the following 25 Figure panels contain new data that has been added to this revised version of the manuscript:

Figure 1D-H
Figure 2E-I
Figure 3D-F
Figure 4A-B
Figure 6C, E
Figure S1
Figure S2E-J
Figure S4A-B

The most important new data that has been added:

- Decreased migration in cells genetically deficient for SLC12A2 or treated with inhibitors to SLC12A2, SLC12A6 and SLC12A7 simultaneously.
- Decreased migration of cells in medium with reduced Na⁺ and/or Cl⁻ concentrations.
- Direct measurement of CCL21-induced ion entry, and its dependence on WNK1.
- Direct measurement of CCL21-induced water entry, and its dependence on WNK1.
- Demonstration of increased diffusion speed of macromolecules at the front of the migrating cell compared to the back, consistent with localized water entry at the front of the cell, and regulation of these rates by WNK1 and water influx.

Please note that due to the addition of the new data, Figure numbering has changed from the original version of the manuscript.

Reviewer #1 (Remarks to the Author):

These investigators previously demonstrated that the migration of T cells depends on the WNK1 kinase, its downstream kinases STK39 and OXSR1 (also called SPAK and OSR1), and the ion transporter regulated by SPAK and OSR1, the sodium-potassium-2-chloride cotransporter SLC12A2 (also called NKCC1). Here, they follow up on those studies to understand the mechanisms by which this kinase/ion transporter pathway may be controlling cell migration. They demonstrate decreased T cell migration in response to CCL21 in ex vivo assays when the activity of WNK1, SPAK/OSR1, or NKCC1, as well as the water channel AQP3, are decreased via genetic or pharmacological manipulations. The activated (phosphorylated) forms of WNK1, OSR1, and NKCC1, as well as total OSR1 and AQP3, are preferentially found at the leading edge of migrating T cells, where they colocalize with CDC42. They hypothesize that ion and water influx through NKCC1 and AQP3, regulated by the WNK-SPAK/OSR1 kinase cascade, increases local ion and water influx to generate space between F-actin and the plasma membrane, in order to add actin to the leading edge F-actin and allow migration via the Brownian ratchet mechanism. This is examined using cell biological and physiological approaches.

Overall, this is an attractive hypothesis that may be relevant not only to T cell migration, but also to other cell types. Additional experiments could further bolster the findings. For example, all the data on NKCC1 and AQP3 rely on a

single pharmacological inhibitor for each transporter/channel. The *Slc12a2* knockout mice are viable, and in fact, the authors used T cells from *Slc12a2* knockouts in their previous paper, and saw effects similar to WNK1 knockdown or SPAK/OSR1 double knockdown, unlike in this paper in which they consistently see partial effects of bumetanide. Although they argue that this could be due to redundant effects of four other SLC12A transporters, no information is provided about which of these are expressed in T cells, nor is any attempt made to experimentally test this. This is important since their model is that water moving through AQP3 follows ion influx through ion transporters.

The reviewer is referring to Fig 7a in the Köchl et al (2016) paper where we used RNAi in Jurkat T cells to carry out a double knockdown of OXSR1 and STK39 or of SLC12A2 and showed decreased CXCL12-induced migration in a Transwell assay, which was similar to a knockdown of WNK1. In the studies in this manuscript, we are working with primary mouse CD4⁺ T cells and measuring the speed of migration in response to CCL21 under agarose, so it's a different system and may not be surprising that the results are quantitatively different. We find that a double mutation in *Oxsr1* and *Stk39* and the inhibition of SLC12A2 give less severe defects than complete loss or inhibition of WNK1. Nonetheless, we agree that it would be important to broaden our analysis of the SLC12A-family of ion co-transporters.

We have added data on the expression of the SLC12A-family members (new Figure S2D). To complement our previous results showing that an SLC12A2 inhibitor causes reduced migration speed, we now show that genetic ablation of SLC12A2 gives a similar small reduction in migration speed (new Figure 1D). In addition, we have added data on migration speed in cells treated with an inhibitor of SLC12A6 and SLC12A7 alone and in combination with an SLC12A2 inhibitor (new Figure 1E). These results show an additive effect of using both inhibitors together, supporting the hypothesis that there is redundancy in function between SLC12A-family members.

Another approach to testing the role of NKCC1 would be to omit extracellular chloride.

As requested, we have measured CCL21-induced migration speed in cells where we reduced the concentration of NaCl thereby removing both Na⁺ and Cl⁻ ions and saw a progressive reduction in speed (new Figure 1G). In addition, we placed cells in medium where just Cl⁻ ions were removed by replacing NaCl with Na gluconate and again saw reduced migration speed (new Figure 1H). These results support our hypothesis that Na⁺ and Cl⁻ ions are important for chemokine-induced T cell migration.

Genetic manipulation of AQP3 would also strengthen the findings using the AQP3 inhibitor, eg using T cells harvested from *Aqp3* knockout mice (which are viable).

Some time ago, based on published papers, we had tried to obtain AQP3-deficient mice, but the labs we approached were unable to provide them. More recently we have identified that there is a EUCOMM allele of *Aqp3* available which can be used to generate *Aqp3* knockout mice. However, the process of importing and establishing this strain and crossing it to a germline Cre driver to generate the knockout mice will take a substantial amount of time and is beyond the scope of what we could practically do for this response. To circumvent this, we attempted to knockout *Aqp3* in primary mouse CD4⁺ T cells using CRISPR gene editing. However, despite many attempts, the best we could achieve was around 30% mutation frequency, which is not good enough to carry out informative migration experiments. Nonetheless, we note that AQP3-deficient T cells have already been shown to have defective chemokine-dependent actin

polymerization defects, as well as decreased migration in Transwells in response to chemokine (Hara-Chikuma et al, 2012) and we cite this paper in our manuscript.

Another question is whether the WNK1-SPAK/OSR1 kinase cascade is regulating AQP3, since AQP3 inhibition fully recapitulates the phenotype observed with loss of WNK1 or SPAK/OSR1. This could be examined in heterologous cells.

The WNK1 pathway regulates AQP3 polarization. We have shown that in polarized migrating T cells, AQP3 is localized to the front of the cell (Figure 3G). When WNK1 is inhibited or inactivated, T cells no longer polarize and there is no polarization of AQP3 (see Figure R1). The same is true for all the other proteins analyzed in Figure 3G. We have not included this data in the paper, but could do so if the reviewer wishes.

Conceivably, AQP3 could be regulated by changes in intracellular trafficking, and this might be altered in cells with defects in the WNK1 pathway. This is difficult to study in primary T cells which are very small and possess very little cytoplasm outside the nucleus. As the reviewer suggests this is an area of research that could be pursued in other cell types. However, although it is an interesting question, we feel these further studies lie outside the scope of the current manuscript.

The authors demonstrate that mild hypotonicity rescues the migration defect in T cells treated with a very low concentration (0.2 μM) of WNK inhibitor (Fig 5C). They get a smaller degree of rescue in isotonic medium with low NaCl (Fig 5D). A known consequence of hypotonic exposure is a decrease in intracellular chloride due to the regulatory volume decrease response. Since chloride is a negative regulator of WNKs, and a very low concentration of WNK inhibitor is being used in this experiment, one possibility is that the hypotonic medium is allowing increased activation of the residual WNK, rather than the water influx

per se explaining the rescue. Thus, converse to the third set of bars in this figure, it would be important to lower osmolality while keeping chloride the same (or even high) to rule out this possibility. This could be done by using L-glucose in the isotonic medium, and omitting it for the hypotonic. Simply lowering extracellular chloride without changing tonicity may not be sufficient to lower intracellular chloride, depending on the chloride conductance of these cells, so the experiment performed (ie isotonic low NaCl) does not rule out a possible intracellular chloride effect. Examining the degree of WNK1 S382 phosphorylation (reflective of WNK autophosphorylation and activation) under the various conditions could also lend insight into this question.

The reviewer suggests that the 20% reduction in the concentration of chloride in the hypotonic medium in Figure 7C, 7D may be activating WNK1, despite the presence of a low dose of WNK inhibitor, and that this accounts for the rescue in migration speed. To test this idea, we needed to analyze WNK1 activity, which we have been doing by measuring levels of p-OXSR1 (see for example Figure S2C). Unfortunately, the current batch of polyclonal, sheep serum-derived anti-p-OXSR1 antibodies does not give reliable detection of the p-OXSR1 protein on immunoblots, and despite numerous attempts, we have not been able to address this point. We have edited the text to refer to the possibility that low chloride may be activating WNK1 (page 16, paragraph 1).

Minor:

1. It's a bit difficult to follow the combination of genetics and bone marrow chimeras by reading the Methods/Results/Figure legends. It would be helpful to have a supplemental table in which this is detailed for the figures using these, ie, genotype, when tamoxifen is applied, which bone marrow cells are transferred into which mice, and where the T cells are harvested from, as well as abbreviations used (eg, "RCE" = ROSA26 CreERT2). Similarly, "CTV" appears in the video/figure legends without being spelled out; readers should not have to go to the Methods to figure out what this non-standard abbreviation stands for. It would be helpful to indicate in Figure 1C that "WNKi" = "WNK inhibitor (WNK463)", and in 1D that "SLC12A2i" = "NKCC1 inhibitor (bumetanide)" and "AQP3i" = "AQP3 inhibitor (DFP00173)."

To address the reviewer's point, we have generated new Figures showing how CD4+ T cells deficient in either WNK1 or OXSR1 and STK39 were made following generation of bone marrow chimeras and subsequent tamoxifen treatment (new Figure S1A-B). Within the legends of these, RCE is defined. In addition, we have explained the procedure by which we generated WNK1-deficient T cells in the Results (page 5, paragraph 2). As requested, CTV and all inhibitors are defined in the legends of all Figures they appear in.

2. The authors note in the discussion that SLC12A2 inhibition reduces migration of glioblastoma cells. In fact, WNK1 or OSR1 knockdown also inhibits glioblastoma cell migration (Zhu, Molecular Cancer 2014, 13:31), further supporting the authors' speculation that the mechanism they describe in T cells may also be relevant to other cell types.

We have added this reference to the Discussion.

Reviewer #2 (Remarks to the Author):

The authors investigate mechanisms by which WNK1 kinase and the cascade it regulates related to SLC12 and water entry through AQP3 creates space for F-actin polymerization at the leading edge of migrating T cells. 3D Motility of tumour cell by blebbing is accepted to require contraction to drive cytoplasm into blebs ahead of F-actin polymerization. There is also a hydraulic engine model for motility that proposes forward movement by osmosis that is secondarily supported by F-actin. That authors propose that water entry through coupled ion transport by SLC12A2 and water transport by AQP3 supports advancing the leading edge ahead of F-actin polymerisation. The authors build a case for this model though a series of experiments focused on polarization of the key molecular players in the cascade, the location of F-actin in relation to the plasma membrane, and manipulation of ions and osmotic gradients. The authors give only passing mention of alternative models that could be quantitatively tested against the authors preferred hypothesis.

1. In Figure 1, how long were the different inhibitors applied. Its striking that WNKi reduced cell volume in a dose dependent manner similar to the effect of the KO, whereas SLC12A2i and AQPi didn't alter cell volume. Were different times of exposure involved.

For the migration assays under agarose shown in Figure 1, cells were injected under agarose containing the inhibitors at least 30 min before imaging was started. For the cell volume measurements shown in Figure 2A-B, cells were treated with inhibitors for 20 min, with CCL21 (or vehicle) for a further 20 min and then volume was measured. Details of timings are listed in the Methods. For each protocol, cells were exposed to all inhibitors for the same amount of time.

The WNKi is more potent at blocking migration. Could concentrations of the WNKi and AQPi that generate similar effects on motility be compared? This may result in a fair comparison that may not support polarization of these signals in relation to the distribution of cytosol.

As suggested by the reviewer, we have titered the dose of WNKi, examining the effect on migration speed and circularity (a measure of polarization) (Figure S2D). We found that 5 μM WNKi resulted in a maximal inhibition of CCL21-induced migration and polarization, 0.5 μM WNKi partially inhibited migration and polarization, whereas 0.2 μM WNK1 reduced migration speed by a small amount, but critically, had no effect on polarization. In many assays we have used these sub-maximal (0.2 and 0.5 μM) doses of WNKi in order to allow us to examine the roles of WNK1 in cells that still polarized, making a fairer comparison with inhibition of SLC12A2 or AQP3 (Figures 1C, 2B, 2F, 4B, 6C, 6E, 7B, 7C-E, S2F, S3C). We have also added data on the effects of WNK1 deficiency as well as the effects of the inhibitors on circularity (new Figures S2E, F, H).

The AQPi inhibit all water channels or just AQP3? If inhibiting all water channels shouldn't the effect be greater?

RNA-seq data shows that mouse CD4+ T cells express AQP3, AQP9 and AQP11 (new Figure 2H). DFP00173 inhibits AQP3, but not AQP7 or AQP9 (Sonntag et al, 2019); we are not aware if DFP00173 inhibits AQP11. Thus, treatment with DFP00173 inhibits AQP3 but leaves AQP9 and maybe AQP11 active. This information has been added to the text.

2. A complication with Figure 2 is that WNK1 and OXSR1 appear excluded from the nucleus, whereas GFP is not, so most of the effect in C may be related to

nuclear exclusion and polarization of cytoplasm. Can the authors use a form of GFP that is excluded from the nucleus as a control?

To address the reviewer's point, we imaged a nuclear-excluded GFP and show that this does not accumulate at the leading edge (new Figure 3D-F). Thus, the accumulation of GFP-WNK1, GFP-OXSR1 and GFP-SLC12A2 at the leading edge is not due to nuclear exclusion of these fusion proteins. It is also not a function of passive enrichment of WNK1 pathway proteins at the leading edge, as the nucleus-excluded GFP localizes to the rear of the cell.

Similarly, for SLC12A2, a membrane protein, GFP is not an appropriate control. The CAAX construct used later might be more suitable, although this is potentially complicated if SLC12A2 is not confined to the plasma membrane.

We confirmed the localization of SLC12A2 and p-SLC12A2 at the leading edge in Figure 3G, H where we image fixed cells and compare to the localization of CDC42 and CD44 that mark the membrane at the leading and trailing edges respectively. In addition, we note that the plasma membrane-tethered CAAX construct referred to by the reviewer, is uniformly distributed around the migrating cell, with no preferential accumulation at the leading edge (Figures 6A, B).

3. Figure 3 interprets a separation of the average F-actin location from the average membrane location as evidence for a water filled gap between F-actin and the membrane. There are two weaknesses of this approach. First, we know that the lymphocyte surface has sub-resolved membrane protrusion that contain F-actin, in addition to cortical f-actin and the extensive branched F-actin network in the leading edge that is expected to have a include F-actin mass that is well separated from the membrane. So the authors would need to consider other models that can generate the same separation and somehow rule these out. The genetics of the system also seems to impact the extent of dendritic nucleation of F-actin and this could be the major effect. The other issue is that building a reasonable model would require information on membrane topology that is not resolved. They would need a good way to go from the averaged data to a model of membrane topology for the lateral and leading-edge structures under different conditions to generate a model. This would require electron tomography or at least SIM level super-resolution data, if not dSTORM or expansion microscopy. The observation that perturbations that alter motility also alter this measurement could have many explanations that don't involve a water filled space into which F-actin polymerizes.

We agree with the reviewer that the observation of an increased distance between the peak of cortical F-actin density and the plasma membrane as measured by iSIM (now Figure 5) does not unequivocally show that there is increased spacing between the membrane and the underlying F-actin. There could, for example, be a wider band of F-actin at the leading edge in cells that are migrating faster, and inhibitors that slow the cells down, may also cause a less wide band of F-actin. We have amended the text to reflect this.

The suggestion to image the F-actin at the leading edge by EM tomography is a good one. We spent several months in collaboration with experts trying this approach as well as STORM imaging, but we were unable to resolve the actin structure in the leading edge of migrating T cells. This is probably because the protruding leading edge is small and the actin cytoskeleton is too dense. Thus, unfortunately, we feel these experiments are beyond the scope of this manuscript.

4. In Fig 4 the system used appears to confirm that inhibiting these pathways (WNK1, SLC12A2 and AQP3) reduce the degree of dendritic F-actin, which based on the relationship of the C-actin to the membrane would reduce the signal in this experiment. I don't think this related to the creation of space, but more to the fraction of F-actin that is in range of the membrane anchored F-acting ligand.

We agree with the reviewer that the MPAct probe reports on the amount of F-actin within 10 nm of the membrane rather than directly reporting on the space (now in Figure 6).

Work from the Sixt lab (Mueller et al, 2017) shows that in migrating cells, a reduced load of the membrane on actin filaments at the leading edge results in rapid polymerization of actin filaments perpendicular to the membrane. These outcompete a network of filaments at shallow angles, thereby generating a region with lower F-actin density proximal to the plasma membrane. We hypothesize that water entry at the front of the cell would contribute to this reduced load and hence to lower membrane proximal F-actin (MPA) at the leading edge.

Our data in Figure 6 test this hypothesis. These show that there is less MPA at the leading edge than at the trailing edge, and that inhibition of WNK1, SLC12A2 or AQP3 causes an increase in the amount of MPA at the leading edge, consistent with this hypothesis. We have ensured that the manuscript text does not state that the MPAct probe directly reports on space between the membrane and the actin cytoskeleton.

5. In Figure 5 the authors apply the approach I suggested above and use the WNK1i at a 0.2 μ M to allow polarization with effects more similar to the SLC12A2 or AQP3 KO/inhibitor studies. However, even with this approach, it is not clear that the results are as they would predict. Reducing tonicity would just non-specifically swell the cells through all aquaporins and this restored motility and makes the motility independent of weak inhibition of WNK1i.

As the reviewer suggested earlier, we used the low dose of WNKi (0.2 μ M) because this partially inhibits migration but does not affect polarization of the cell, and more closely resembles the effect of inhibition of SLC12A2 or AQP3. Under these conditions, hypotonic medium rescues the migration defect (now Figure 7C-D). It also rescues actin flow rates (Figure 7E) and restores reduced MPA at the leading edge (Figure 6E).

The reviewer points out that the hypotonic treatment may cause water entry all around the cell and not just at the leading edge. This may be the case, though we note that AQP3 is polarized to the leading edge, so more water may enter at the leading edge during hypotonic treatment. In any case, we propose that it is only at the leading edge that water entry would be able to promote migration, since this is where the motility machinery of actin polymerization is polarized in response to chemokine receptor signaling. Our hypothesis is that the water entry would facilitate actin polymerization at the leading edge, not that it would stimulate actin polymerization all around the cell.

To further support our hypothesis that WNK1 activation at the leading edge causes localized water entry we transfected T cells with Qdots, brightly fluorescent nanocrystals, and measured their diffusion rates. We found that Qdots diffused faster at the front of the cell compared to the back, consistent with localized water entry at the front leading to a less dense cytoplasm (new Figure 4). Furthermore, the diffusion of Qdots was slowed down by partial inhibition of WNKi, and this was rescued by putting the cells into hypotonic medium. Importantly, Qdot diffusion speeds were faster at the leading edge than in the rear of the cell even in the hypotonic-rescued cells,

consistent with continued polarization of water influx even in the hypotonic rescue of low-dose WNKi-treated cells.

What happens if a higher concentration of WNKi or the knockout? Does the low tonicity rescue motility and F-actin flow?

At high doses (e.g. 5 μ M WNKi) the cells no longer polarize or migrate and hypotonic medium does not rescue migration. We speculate that WNK1 may be playing other roles in the cell, including contributing to the formation of a polarized morphology.

Are there situations where effects are reverses and WNKi stimulations mobility?

We have never seen WNK1 inhibition increasing motility at any dose (see Figure S2D).

I worry that there are other biophysical and biochemical effects that lead to collapse of polarity with WNK1i, knockout and its restoration by manipulation of cytoplasmic ion levels. Certainly, effect on the order seen with AQPi could be based on the proposed water entry promoting membrane protrusion, but the entire effect of WNK1 loss or inhibition seems well beyond what can be accounted for by any osmotic effects.

We agree with the reviewer. At a high dose of WNK inhibitor, or in WNK1-deficient cells there is a complete loss of polarization of the cells and this cannot be reversed with hypotonic medium. We agree that WNK1 must additionally be doing something other than regulating water entry at the leading edge. We have amended the Discussion to make this point clear.

It should also not be forgotten that WNK1 deficient cells become hyperadhesive, from earlier work from this lab, which also requires actin-based contractility to activate LFA-1. How does this relate to this model? Can the cells spread and generate F-actin based contractility in the interface? There are also NaCl dependent kinases in the cells that could also provide a mechanism for effects of NaCl entry.

In the Köchl et al (2016) paper we investigated the relationship between the increased adhesion and reduced migration of WNK1-deficient T cells. It was possible that the increased adhesion could be responsible for the reduced rate of migration. However, we were able to show that this was not true. If we blocked integrin adhesion, the cells slowed down even further (Fig 4c of Köchl et al), demonstrating that the adhesive and migratory phenotypes are distinct. Furthermore, in this earlier publication we showed knockout or knockdown of SLC12A2 reduced migration but had no effect on adhesion (Fig 6a, b, d of Köchl et al), again supporting the view that the adhesive and migratory phenotypes of WNK1-deficient T cells are distinct phenotypes. This point is discussed in the Introduction of the current manuscript (page 4, paragraph 2).

Reviewer #3 (Remarks to the Author):

The manuscript by de Boer et al. builds on a previous study by the authors on the identification via an RNAi screen of WNK1 as a negative regulator of integrin-mediated adhesion and a positive regulator of chemokine-driven migration in T cells (Köchl R. et al., Nat Immunol 2017). As it is the case in other cellular models, the authors had found that downstream of the chemokine receptor CCR7, WNK1 activates the OXSR1 and STK39 kinases and the SLC12A2 ion co-transporter.

These previous data were suggestive of the fact that ion influx is required for T cell migration.

In the current study, the authors propose to investigate how the WNK1-SLS12A2 axis regulates T cell migration. They report that deficiency/inhibition or expression of mutant of WNK1, OXSR1, STK39 and SLS12A2 resulted in decreased migration of murine CD4⁺ T cells under confinement (under-agarose, 3D collagen-I matrix). This piece of data consolidates data from their previous study. They then show the polarization of WNK1, OXSR1 and SLS12A2 at the leading edge of T cells migrating in confined in vitro settings. This suggests that CCR7 signaling via WNK1, OXSR1 and STK39 leads to localized entry of Na⁺, K⁺ and Cl⁻ ions via SLC12A2 at the leading edge of migrating T cells. The authors then report that CCL21 induced a volume increase in WNK1-expressing T cells, whereas no such increase was seen in the absence of WNK1, or if WNK1 was inhibited. This supports the hypothesis that WNK1 pathway activity results in water influx, which is required for volume regulation. As they report, this hypothesis aligns with the concept of the « osmotic engine model » (Stroka, K.M. et al. Cell 2014). Then, the authors focus on AQP3 as a candidate aquaporin downstream the WNK1-SLS12A2 axis. In line with the hypothesis of the authors, AQP3 inhibition caused a significant decrease in the speed of CCL21-induced migration of T cells under agarose or in a collagen-I matrix. The authors then employ structured illumination microscopy and previously validated fluorescent probes to explore the impact of the WNK1-SLS12A2-AQP3 axis on the spacing between the leading edge plasma membrane and the underlying actin cytoskeleton. Treatment of T cells with inhibitors targeting WNK, SLC12A2 or AQP3 resulted in an increase in the membrane-proximal actin distance at the leading edge, implying that this axis via polarized osmotic uptake of water might favor the actin polymerization process driving cell protrusion and migration. To complement these observations, the authors measure retrograde actin flow to show that it is reduced in WNK1-deficient T cells or T cells treated with inhibitors of WNK1, SLC12A2 or AQP3. Finally in order to more directly test the contribution of water influx in the observed process, they modulate the tonicity of the medium to report increased migration speed in cells where WNK1 had been partially inhibited.

Overall this is an interesting in vitro study addressing very basic aspects of polarized T cell migration. Although the investigated pathway is not novel, the study provides some new exploration to connect chemokine sensing to polarized migration via a conserved pathway involving ion transport and osmotic regulation. The manuscript is well written and the data are presented in clearly organized figures. Some of the conclusions that the authors formulate are however too far stretched as they go beyond what is demonstrated through the provided experimental work. As explained below, further work would be required to properly address some of the key points the authors try to make in this study.

Major points:

1- Possibility that influx of H₂O₂ rather than (or together with) H₂O would explain the role of the WNK1-SLC12A2-AQP3 axis in regulating leading-edge actin remodeling to drive polarized T cell migration.

The current study agrees with a previous publication reporting that AQP3-deficient T cells have defective migration in response to various chemokines both in vitro and in vivo (Hara-Chikuma, M. et al. Chemokine-dependent T cell

migration requires aquaporin-3-mediated hydrogen peroxide uptake. *J Exp Med* 209, 1743-52, 2012). This previous publication however showed a major role of AQP3 for hydrogen peroxide uptake, which was found to activate Cdc42 and polarized actin remodeling via WASP and ARP2/3. Since the authors defend here an alternative mechanism (water influx facilitating actin remodeling by creating space), it is essential that they decipher the relative contribution of water versus H₂O₂ driven mechanisms underlying the phenotype of AQP3-deficient or AQP3-inhibited T cells. The authors would need to inhibit H₂O₂ transport to study the contribution of water influx to the phenotype (migration and leading edge actin remodeling). It is also expected that the authors introduce what the study by Hara-Chikuma, M. et al. showed and discuss the relative H₂O/H₂O₂ contribution.

As requested, we have investigated the role of H₂O₂ in T cell migration, since AQP3 can transport H₂O₂ as well as water. Using catalase to eliminate H₂O₂, we show that H₂O₂ is not required for CCL21-induced migration of mouse CD4⁺ T cells (new Figure S4B). It is not clear why our results are different from those reported in Hara-Chikuma et al 2012. There are a number of methodological differences. We measured migration speed in response to CCL21 under agarose, whereas Hara-Chikuma et al measured % migration through a Transwell in response to CXCL12. In addition, we used 20 U/ml catalase whereas Hara-Chikuma et al used 2000 U/ml catalase. Our choice of 20 U/ml was based on a titration where we investigated how much catalase is needed to eliminate 1 mM H₂O₂ as detected by DCFDA fluorescence. We found that 20 U/ml was sufficient for this purpose, with no further improvement seen at 200 U/ml (new Figure S4A). Thus, we chose 20 U/ml catalase, since this was the lowest concentration that was sufficient to eliminate 1 mM H₂O₂. It is possible that the supramaximal 2000 U/ml catalase used by Hara-Chikuma et al may have had off-target effects that compromised migration. In any case, the hypotonic rescue experiments shown in Figure 7 demonstrate that water entry is able to reverse the migration defects in WNK1-inhibited cells, supporting the importance of water flux in T cell migration.

2- Measuring ion transport and water influx

While the reported experiments (deletion or inhibition of SLC12A2 and AQP3, use of hypotonic medium) are suggestive of a contribution of ion transport and water influx in promoting actin remodeling of T cells stimulated with CCL21, direct measurements of such fluxes are missing. Such measurements would be quite important to sustain the claims made by authors in their manuscript.

We agree with the reviewer that it would be important to directly demonstrate CCL21-induced ion and water flux. We used inductively-coupled plasma mass spectrometry to measure ion content in T cells and found the CCL21 stimulation leads to an increase in K⁺ ions in the cell which was blocked when WNK1 was inhibited (new Figure 1F). We were unable to get reliable data for Na⁺ or Cl⁻ because the concentration of these ions is high in the extracellular medium resulting in a high background of these ions in the samples, precluding accurate measurements of these ions.

In addition, we directly measured water uptake using deuterated water (²H₂O) and then NMR to quantify how much ²H₂O had been taken up by the cells. Again, we found that CCL21 stimulation leads to an increase in water uptake, which is blocked by WNK inhibition (new Figure 2G), closely matching the volume changes resulting from CCL21 stimulation and WNK1 inhibition.

In relation to this comment, the following sentences need to be revised according to what is precisely demonstrated experimentally:

“chemokine receptor-induced activation of the WNK1 pathway leads to water influx, most likely via AQP3, which is required for actin polymerization and migration”

“this chemokine receptor-induced water entry generates increased spacing between the plasma membrane and the underlying actin cytoskeleton at the leading edge”

“WNK1 activity and ion and water influx are required to maintain reduced MPA at the leading edge of T cells”

“Thus, WNK1 and ion and water influx are required for actin retrograde flow”

All of these sentences have been edited to ensure that they accurately represent our findings. In addition, we have edited other parts of the text in a similar way to make sure that we specify our results precisely.

3- Resolution of the F-actin-membrane distance measurements

The authors claim they want to investigate the fluctuating gap separating the membrane and the actin filaments, which is a 10nm range phenomenon, with structured illumination microscopy. However, such approach is not expected to provide a sufficient resolution. The plasma membrane to “peak” F-actin signal that the authors measure seem to correspond to the distance between the plasma membrane and an inner part of the lamellipodial protrusion (for leading edge measurement) or to the core of the cortical actin (for the cell side measurements). While the reported measurements seem to indicate an effect the tested inhibitions, they might only reflect an overall change in actin cytoskeleton organization. Differently, the MPAct/CaaX ratio measurements appear to be a more adapted method to measure of the distribution of membrane-proximal actin. I would therefore advise to focus on this latter approach.

In addition to the resolution issue related to the experiments conducted with SIM, the images presented in Figure 3 do not display the expected distribution of F-actin at the polarized T cell leading edge, nor the expected organization of cortical actin on the cell periphery. The staining appears to be too weak and very patchy. The penetrations of LifeAct staining across the membrane-orange staining are also unexpected. It would be important for the authors to validate the quality of the stainings, e.g. by costaining fixed samples with Phalloidin.

The patchy appearance of the images, including the apparent penetration of LifeAct signal through the membrane-orange staining, is due to use of a high resolution imaging method (iSIM) and high speed of acquisition with a 200 ms exposure every 250 ms. Thus, these images will never look like confocal images of fixed cells stained with phalloidin. The analysis of fixed cells is unlikely to help with this, since there is a danger that fixation may change the morphology of the structures being imaged at sub- μ m resolution.

4- Impact of the WNK1-SLC12A2-AQP3 axis on directional migration along chemokine gradients?

While the provided experiments provide solid evidence for the role of the WNK1-SLC12A2-AQP3 axis in CCL21-evoked T cell migration, they do not directly address directional migration along chemokine gradients. It is felt that cell tracks, cell shape and actin measurements in the context of chemokine

gradients would reinforce the current study and would help understanding the precise contribution of this axis to chemokine sensing versus cell polarization version orientation along gradients.

As requested, we have added analysis of the role of WNK1 in directional migration (chemotaxis). Using activated T cells migrating through a collagen-I matrix in response to a gradient of CXCL12, we were able to show that WNK1 is required for their directional migration (new Figure S2J).

5- Which T cell motility steps are primary governed by the WNK1-SLC12A2-AQP3 axis ?

The authors had previously reported that WNK1 is activated downstream of CCR7, LFA-1 and the TCR (Köchl R. et al., Nat Immunol 2017). A question of central physiological relevance is therefore whether the WNK1-SLC12A2-AQP3 axis controls to a similar degree steps of T cell migration known to be distinctly balanced by chemokine receptors, integrins and the TCR, namely trans-endothelial migration, interstitial migration and antigen-presenting cell scanning. Complementary in vivo and in vitro experiments would help addressing this point. In particular, it is well accepted that CCR7 and the TCR play rather opposite roles in controlling T cell motility (polarized migration versus stop of migration). The authors could try unraveling how the addition of a TCR trigger to that of CCR7 alters the degree of WNK1 activation and possibly the downstream axis.

Most of the questions raised by the reviewer were addressed in our earlier publication (Köchl et al, 2016). In this earlier study we analyzed the in vivo migration of WNK1-deficient and control T cells and showed they entered lymph nodes less efficiently, being more likely to be associated with lumen and perivascular areas of the lymph node, they adhered more strongly to endothelium, but migrated more slowly on it and transmigrated more slowly through it (Fig 3a-g). Using intra-vital imaging, we showed that they migrated more slowly in lymph nodes once they have entered them (Fig 4a-d). Furthermore, we showed that the decreased rate of migration of the WNK1-deficient cells was not due to their hyperadhesive integrins. Blocking integrin binding in vivo caused the WNK1-deficient T cells to slow down even further, showing that if anything, the cells were using their hyperadhesive integrins to promote their migration (Fig 4c). We refer to these results in the Introduction. We feel that further studies, e.g. on the efficiency of scanning APCs, although interesting, is outside the scope of the current study.

6- WNK1-SLC12A2-AQP3 axis in T cell subsets?

Another direction that could greatly raise the interest for a larger community of immunologists would be to assess protein expression related to the explored pathway (WNK1, OXSR1, STK39, SLC12A2, AQP3) in various T cell subsets and under distinct activation/differentiation conditions. Indeed various T cell subsets (Th1 vs Th2, effector T cells vs T regulatory cells, resting versus activated T cells) display distinct motility properties. While there has been an obvious focus on trying to relate this to the expression of specific chemokine receptors and adhesion receptors, much less is known on the possible tuning of the conserved intracellular pathways that govern motility, such as the one studied here. Therefore protein expression study completed with a few key functional assays with drugs (as done on naïve CD4+ T cells in the manuscript) might reveal very interesting aspects of motility regulation underlying the properties of T cell subsets.

We have added data showing that WNK1 is required for the migration of activated T cells (new Figure S2I-J). We have previously shown that WNK1 is required for the migration and lymph node homing of CD8⁺ T cells (Köchler et al 2016), for the CXCL12-induced migration of CD4⁺CD8⁻ thymocytes (Köchler et al 2020) and more recently for the CXCL13-induced migration of B cells in vitro and in vivo (Hayward et al 2023). We have cited these studies in the manuscript. Thus, a picture is emerging that this pathway is of importance for the migration of many immune (and non-immune cell types). Further studies on the role of the WNK pathway in the migration of different T cell subsets (Th1, Th2, Th17 and Treg) would be an interesting area to pursue, but we feel these are outside the scope of the current study.

Nonetheless, inspection of a published RNA-seq dataset on CD4⁺ T cell Th subsets, which includes naive, Th1, Th2, Th17, iTreg and Treg cells (Th-Express, <https://th-express.org/>) shows some interesting changes in the expression of some members of the Wnk1 pathway:

Gene	naive	Th1	Th2	Th17	iTreg	Treg
Wnk1	2720	2620	1250	1240	519	1740
Wnk2	1.18	90.1	0	3.9	4.85	0.22
Wnk3	5.44	0.146	0.433	1.69	1.24	3.96
Wnk4	62.4	5.43	12.5	6.66	18.2	39.5
Oxsr1	615	670	463	809	207	453
Stk39	294	732	459	250	599	320
Slc12a1	0	0	0.0824	1.84	0	0.327
Slc12a2	55	78.8	83.7	546	147	106
Slc12a3	8.67	2.21	1.61	14.9	0	6.48
Slc12a4	345	252	272	268	507	297
Slc12a5	13.5	1.7	25.7	3.92	1.81	21.4
Slc12a6	502	332	400	211	211	598
Slc12a7	3650	704	694	337	583	1210
Slc12a8	17	49.4	5.4	21.6	5.49	38.9
Slc12a9	341	132	315	110	230	606
Aqp1	0	0.175	0.161	0	0	33.4
Aqp2	0.447	0	0	0.356	0	0
Aqp3	105	530	607	4560	2570	89.7
Aqp4	0.171	0	0	1.99	0	1.81
Aqp5	0.409	0	0	0.165	0	1
Aqp6	0.269	0	0	0	0	0
Aqp7	21.9	1.69	26.7	5.12	0.758	13.9
Aqp8	0.213	0.309	0	0	0	0
Aqp9	4.73	1360	935	218	31.4	8.85
Aqp11	36.1	6.43	38.4	6.23	12.2	31.4
Aqp12	2.33	0	0	0	0	0

This shows that for all subsets examined, Wnk1 is the most abundantly expressed WNK kinase. Oxsr1 is more abundant than Stk39 in all subsets except in iTreg and Slc12a7 is the most abundant of the ion co-transporters in all subsets except Th17 where Slc12a2 is expressed at a higher level. The most variation is seen in the Aqp family members. Whereas Aqp3 is expressed at the highest levels in naive, Th17, iTreg and Treg cells, Aqp9 is dominant isoform in Th1 and Th2 cells. Overall, all CD4 subsets express at least one member of each of these pathway proteins, but isoform usage varies. Future studies could be aimed at understanding the meaning of these differences.

Further points requiring clarification (from figures):

Figure 1H: increase in cell vol upon CCL21 is very limited. It is difficult if not impossible to assess effects (or lack of effect) of inhibitors under such settings.

We agree that the fold changes in volume in response to CCL21 are not large, and vary between experiments. This is an inherent feature of the CASY cell counter method used in these panels for volume determination. In addition, the method used in the CASY cell counter, electrical conductance, does not give accurate absolute volume measurements, but is more suited to relative volume changes. In view of this, we have changed the graphs in Figure 2A-D to show volumes normalized the mean of control cells in the absence of CCL21 and inhibitors. Thus, these data show the relative fold changes in volume in response to CCL21 and the effects of inhibitors on this.

To extend and further validate the CCL21-induced volume changes we set up an independent method for volume measurement by imaging live CD4+ T cells on ICAM-1 coated microscope slides \pm CCL21 stimulation. We used T cells expressing a plasma membrane-tethered tdTomato to allow us to visualize the edges of the cell, and then used high resolution instant structural illumination microscopy (iSIM) microscopy to acquire 3D images of the cells. Absolute volumes were calculated from reconstructions of the cells. The results show a clear CCL21-induced increase in volume which is abrogated by treatment with the WNK inhibitor (new Figure 2E-F).

Figure 2B,C,E: indicate nb of cells (to make it clearer that this comes from cell averaging). Why are C and E represented differently (where is dispersion of individual cells)?

The numbers of cells used in these figure panels (now Figure 3C, F, H) are listed in the legend. We averaged ≥ 1244 cells in 3C, 243 cells in 3F and ≥ 47 cells in 3H. The text of the legend has been edited to make it clear that the graphs represent averaged data. The graphs in 3C, 3F and 3H look different in part because the data in 3C and 3F was determined along the length of the cell in μm , imaging every $\sim 0.3 \mu\text{m}$, whereas in 3H the data was binned into 12 bins along the length of the cell, each representing 1/12th of the cell length. We have now added the individual data points in the graphs in 3C and 3F to make them more closely resemble the graphs in 3H. In addition, all the graphs show the mean as a solid line and $\pm 95\%$ confidence interval as a shaded region. In the graphs in 3C and 3F the data is from a much larger number of cells than the data in 3H, which makes the 95% confidence interval so small that it is not visible around the line. The legend has been edited to make these points clear.

Figure 3 : normalization of intensity values: in B both membrane and actin seem to be normalized with max value = 1. In C only membrane seems to be normalized.

In all cases, for each individual cell, the peak of the membrane fluorescence and the peak of the LifeAct-eGFP fluorescence was set to 1. In Figure 5B (new numbering) data is shown for just one example cell, hence both curves reach a peak of 1. In Figure 5C we have averaged data from ≥ 23 cells and then aligned the data such that the position of the peak of membrane fluorescence was set to 0 nm on the x-axis. This makes the normalized membrane fluorescence reach 1, since the peaks of the membrane fluorescence of all the individual cells are in the same position. However, for the LifeAct-eGFP signal, the position of the peak of fluorescence varies in its distance from the membrane peak from cell to cell. When this data is averaged over many cells, this makes the mean peak LifeAct-eGFP signal less than 1.

Figure 4: There is experimental variability of MP Act/Caax profiles along the cell length for vehicle when one compares C and D panels. This variability seems to be as important as the variations noted between the test conditions (WNKi,

SLC12A2i, AQP3i). Please provide further evidence for the robustness of this assay and for the solidity of the data.

The reviewer is correct to note that the graphs in Figure 6C and 6D (new numbering) look somewhat different. The data in these two panels came from separate experiments (at least 2 experiments for each panel) and the differences reflect experimental variation. Importantly, for each panel the different conditions (\pm inhibitor for example) were carried out at the same time, so the data within each panel is directly comparable. Statistical analysis with 2-way ANOVA on this data shows a highly significant FDR-adjusted q-value of < 0.0001 . Thus, we reject the null hypothesis and it is very likely that the inhibitors are having an effect on the distribution of membrane-proximal actin in these migrating T cells, increasing its levels at the leading edge.

REVIEWER COMMENTS

Reviewer #1 (Remarks to the Author):

In this revised manuscript, de Boer et al add new experiments examining K abundance at baseline and in stimulated T cells, and the effects of ion substitutions on T cell migration speed, solidifying conclusions regarding ion influx and cell migration. Similarly, new experiments measuring cell volume using an independent assay (imaging vs. the originally used Coulter counter) and heavy water uptake measured by NMR bolster conclusions on water influx. Additional data on the effect of AQP inhibitors also adds to this understanding. A possible role for H₂O₂, which can also permeate AQP3, is investigated but does not appear to play a role. The experiment looking at Qdot diffusion lends further support to the idea that water influx occurs at the leading edge in a WNK-dependent manner, as does the reversal of the WNK inhibitor effect with hypotonic medium (in this experiment as well as in experiments looking at membrane proximal actin and actin retrograde flow). Overall, the additional experiments have further strengthened this very interesting paper linking WNK1 activation to leading edge ion and water influx and T cell migration.

Minor:

I think the data in Figure R1 are quite interesting and would be worth including, particularly if the authors have all the data already, unless they think they may put the figures in a future paper. Defer to the authors.

The language around low intracellular chloride possibly activating WNK in the revised manuscript is fine. Of note, we have been using an anti-pSPAK/pOSR1 antibody (most recently from Millipore, cat # 07-2273) for many years now, which works very well for Westerns. It will detect both phosphorylated SPAK and OSR1 on the Ser373/Ser325 residue, but that's fine for looking at WNK activity. I am not suggesting this experiment needs to be done for this manuscript, this is just a suggestion that may be helpful in ongoing work.

Supplementary Figure 1 is very helpful for understanding the genetics, and the addition of inhibitor information into the figure legends is also helpful.

Many investigators know the SLC12 transporters better by their common names (eg, NKCC1, KCC1) than by which SLC12 is which – I had to look that up. Would be helpful to add to the Fig. S2 legend, “SLC12A2 (NKCC1), SLC12A4 (KCC1), SLC12A6 (KCC3), SLC12A7 (KCC4), and SLC12A9 (CIP) are expressed.”

On the face of it, the KCC inhibitor experiments in Figure 1E do not make much sense with the model. Inhibiting KCCs typically decreases ion (K and Cl) efflux from cells, which would be expected to increase cellular water. However, the inhibitory effects of the KCC inhibitors could be consistent with a model in which water first enters (via NKCC1 and AQP3 +/- other transporters) and then exits (via KCCs) to allow repeating cycles of the “facilitated Brownian ratchet.” This requires more testing, however, and I think is beyond the scope of this paper (though may be interesting to pursue in future). DIOA is also not the cleanest of drugs, so this would require some additional genetic experiments and/or use of newer inhibitors with more specificity. I still object to the idea that KCCs are “compensating” for NKCC inhibition (since these transporters usually transport ions in opposite directions, and NKCCs are activated by WNK-SPAK/OSR1 signaling, whereas KCCs are inhibited, thus NKCC1 and

KCC inhibitors may be expected to cancel each other out rather than having additive effects). I would consider removing the new sentence in lines 441-443 of the discussion (+/- removal of Figure 1E, saving it for a later day once the role of KCCs in the process are better elucidated), although since this is a very small part of the paper I will defer to the authors' judgment on this point.

Reviewer #2 (Remarks to the Author):

The authors have made a strong case for a role of WNK1 regulated ion and water transport as part of what WNK1 does to promote motility. A number of new experiments were performed to support the model. There is only one new experiment where I'm concerned about the result. In the Qdot "diffusion" experiment the result is reported as a speed ($\mu\text{m/s}$) rather than a diffusion coefficient ($\mu\text{m}^2/\text{s}$). Could they calculate a diffusion coefficient and perform an analysis of mean sq displacement vs time to test for simple diffusion? The differences in apparent speed may be due to the differences in the geometry to the 3D compartment accessible to the Dots in the different locations in relation to the depth of field of the imaging system. Their speed measurement may also be influenced by non-diffusive components such as transport. The transport speed of the F-actin network in the lamellipodium could reach several hundred $\mu\text{m/s}$, such that the result could be influenced by different transport effects that should be picked up in a standard analysis of particle motion. So even the relative difference in the leading edge that supports the model could be impacted by the F-actin flow that would be a target of the proposed effect. This experiment would be the topic of an entire Biophys J paper and I don't know if they want to introduce all the issues associated with this here at this stage. My suggestion would be not to include it at this point, although it's a cool idea.

Reviewer #3 (Remarks to the Author):

The authors have invested in a number of novel experiments to reinforce their manuscript along the suggestions of the reviewers. Most of the points I had initially raised have been addressed in a satisfactory way.

Point 1 (relative H₂O/H₂O₂ contribution): new data with catalase inhibition of H₂O₂ have been added, along with a titration experiment.

Point 2 (measurements of ion and water flux): new data of K⁺ ions and deuterated water have been added.

Point 3 (issue about F-actin to membrane distance analysis by SIM): the authors have not directly addressed my point but did in their answer to point 3 of reviewer 2. They were not able to provide new data with higher resolution microscopy approaches although they attempted to employ EM tomography and STORM approaches with experts. They at least amended the text to explain the limitations of the SIM approach.

Point 4 (increasing physiological relevance by studying chemotaxis): new data with a gradient of CXCL12 were added.

Point 5 (increasing physiological relevance by various motility steps): the authors referred to

their previous in vivo study in which the role of WNK1 was studied for the steps of transendothelial migration and interstitial migration. They considered that investigating the antigen-presenting cell scanning activity would be outside the scope of the manuscript.

Point 6 (increasing physiological relevance by studying various T cell subsets): new data with activated T cells were added. An analysis of published RNA-seq data was provided to the attention of the reviewers indicating that within the Wnk1 pathway, variations in expression and isoform usage is present across various CD4⁺ Th subsets. The authors considered that investigating this further should be part of another study.

RESPONSE TO REVIEWERS' COMMENTS

Reviewer #1 (Remarks to the Author):

In this revised manuscript, de Boer et al add new experiments examining K abundance at baseline and in stimulated T cells, and the effects of ion substitutions on T cell migration speed, solidifying conclusions regarding ion influx and cell migration. Similarly, new experiments measuring cell volume using an independent assay (imaging vs. the originally used Coulter counter) and heavy water uptake measured by NMR bolster conclusions on water influx. Additional data on the effect of AQP inhibitors also adds to this understanding. A possible role for H₂O₂, which can also permeate AQP3, is investigated but does not appear to play a role. The experiment looking at Qdot diffusion lends further support to the idea that water influx occurs at the leading edge in a WNK-dependent manner, as does the reversal of the WNK inhibitor effect with hypotonic medium (in this experiment as well as in experiments looking at membrane proximal actin and actin retrograde flow). Overall, the additional experiments have further strengthened this very interesting paper linking WNK1 activation to leading edge ion and water influx and T cell migration.

Minor:

I think the data in Figure R1 are quite interesting and would be worth including, particularly if the authors have all the data already, unless they think they may put the figures in a future paper. Defer to the authors.

As suggested, we have included this data, along with analysis of the polarization of other WNK pathway proteins in a new Supplementary Figure 6.

The language around low intracellular chloride possibly activating WNK in the revised manuscript is fine. Of note, we have been using an anti-pSPAK/pOSR1 antibody (most recently from Millipore, cat # 07-2273) for many years now, which works very well for Westerns. It will detect both phosphorylated SPAK and OSR1 on the Ser373/Ser325 residue, but that's fine for looking at WNK activity. I am not suggesting this experiment needs to be done for this manuscript, this is just a suggestion that may be helpful in ongoing work.

We thank the reviewer for this suggestion.

Supplementary Figure 1 is very helpful for understanding the genetics, and the addition of inhibitor information into the figure legends is also helpful.

Many investigators know the SLC12 transporters better by their common names (eg, NKCC1, KCC1) than by which SLC12 is which – I had to look that up. Would be helpful to add to the Fig. S2 legend, “SLC12A2 (NKCC1), SLC12A4 (KCC1), SLC12A6 (KCC3), SLC12A7 (KCC4), and SLC12A9 (CIP) are expressed.”

As requested, we have added the common names of the SLC12A-family proteins to the legend of Supplementary Figure 2G.

On the face of it, the KCC inhibitor experiments in Figure 1E do not make much sense with the model. Inhibiting KCCs typically decreases ion (K and Cl) efflux from cells, which would be expected to increase cellular water. However, the

inhibitory effects of the KCC inhibitors could be consistent with a model in which water first enters (via NKCC1 and AQP3 +/- other transporters) and then exits (via KCCs) to allow repeating cycles of the “facilitated Brownian ratchet.” This requires more testing, however, and I think is beyond the scope of this paper (though may be interesting to pursue in future). DIOA is also not the cleanest of drugs, so this would require some additional genetic experiments and/or use of newer inhibitors with more specificity. I still object to the idea that KCCs are “compensating” for NKCC inhibition (since these transporters usually transport ions in opposite directions, and NKCCs are activated by WNK-SPAK/OSR1 signaling, whereas KCCs are inhibited, thus NKCC1 and KCC inhibitors may be expected to cancel each other out rather than having additive effects). I would consider removing the new sentence in lines 441-443 of the discussion (+/- removal of Figure 1E, saving it for a later day once the role of KCCs in the process are better elucidated), although since this is a very small part of the paper I will defer to the authors’ judgment on this point.

As suggested, we have removed the data where we used DIOA to inhibit SLC12A6 and SLC12A7 in the last two columns of Figure 1E. We have taken out all references to 'compensation' between SLC12A2 (NKCC1) and SLC12A6 (KCC3) and SLC12A7 (KCC4). We agree with the reviewer that such phrasing is confusing since the NKCC and KCC proteins have opposing functions (influx v efflux of ions).

Reviewer #2 (Remarks to the Author):

The authors have made a strong case for a role of WNK1 regulated ion and water transport as part of what WNK1 does to promote motility. A number of new experiments were performed to support the model. There is only one new experiments where I'm concerned about the result. In the Qdot "diffusion" experiment the result is reported as a speed ($\mu\text{m/s}$) rather than a diffusion coefficient ($\mu\text{m}^2/\text{s}$). Could they calculate a diffusion coefficient and perform an analysis of mean sq displacement vs time to test for simple diffusion? The differences in apparent speed may be due to the differences in the geometry to the 3D compartment accessible to the Dots in the different locations in relation to the depth of field of the imaging system. They speed measurement may also be influenced by non-diffusive components such as transport. The transport speed of the F-actin network in the lamellipodium could reach several hundred $\mu\text{m/s}$, such that the result could be influence by different in transport effect that should be picked up in a standard analysis of particle motion. So even the relative difference in the leading edge that support the model could be impacted by the F-actin flow that would be a target of the proposed effect. This experiment would could be the topic of an entire Biophys J paper and I don't know if they want to introduce all the issues associated with this here at this stage. My suggestion would be not to include it at this point, although its a cool idea.

We thank the reviewer for these useful points. As suggested, we have removed Figure 4.

Reviewer #3 (Remarks to the Author):

The authors have invested in a number of novel experiments to reinforce their manuscript along the suggestions of the reviewers. Most of the points I had initially raised have been addressed in a satisfactory way.

Point 1 (relative H₂O/H₂O₂ contribution): new data with catalase inhibition of H₂O₂ have been added, along with a titration experiment.

Point 2 (measurements of ion and water flux): new data of K⁺ ions and deuterated water have been added.

Point 3 (issue about F-actin to membrane distance analysis by SIM): the authors have not directly addressed my point but did it their answer to point 3 of reviewer 2. They were not able to provide new data with higher resolution microscopy approaches although they attempted to employ EM tomography and STORM approaches with experts. They at least amended the text to explain the limitations of the SIM approach.

Point 4 (increasing physiological relevance by studying chemotaxis): new data with a gradient of CXCL12 were added.

Point 5 (increasing physiological relevance by various motility steps): the authors referred to their previous in vivo study in which the role of WNK1 was studied for the steps of transendothelial migration and interstitial migration. They considered that investigating the antigen-presenting cell scanning activity would be outside the scope of the manuscript.

Point 6 (increasing physiological relevance by studying various T cell subsets): new data with activated T cells were added. An analysis of published RNA-seq data was provided to the attention of the reviewers indicating that within the Wnk1 pathway, variations in expression and isoform usage is present across various CD4⁺ Th subsets. The authors considered that investigating this further should be part of another study.

There are no further points to address from this reviewer.

REVIEWERS' COMMENTS

Reviewer #1 (Remarks to the Author):

The authors have made the suggested changes. I have no further suggestions.

Reviewer #2 (Remarks to the Author):

I support publication without the data I had questioned. I appreciate the authors responsiveness and have no further concerns with work. It will be a provocative paper for the actin field, but presents a good case for an osmotic boost for fast protrusions.

RESPONSE TO REVIEWERS' COMMENTS

Reviewer #1 (Remarks to the Author):

The authors have made the suggested changes. I have no further suggestions.

Reviewer #2 (Remarks to the Author):

I support publication without the data I had questioned. I appreciate the authors responsiveness and have no further concerns with work. It will be a provocative paper for the actin field, but presents a good case for an osmotic boost for fast protrusions.

There are no further points to address from either reviewer.